# A Robust Non-Clairvoyant Dynamic Mechanism for Contextual Auctions

**Yuan Deng**
Duke University
Durham, NC
ericdy@cs.duke.edu

**Sébastien Lahaie**
Google Research
New York City, NY
slahaie@google.com

**Vahab Mirrokni**
Google Research
New York City, NY
mirrokni@google.com

## Abstract

Dynamic mechanisms offer powerful techniques to improve on both revenue and efficiency by linking sequential auctions using state information, but these techniques rely on exact distributional information of the buyers' valuations (present and future), which limits their use in learning settings. In this paper, we consider the problem of contextual auctions where the seller gradually learns a model of the buyer's valuation as a function of the context (e.g., item features) and seeks a pricing policy that optimizes revenue. Building on the concept of a bank account mechanism—a special class of dynamic mechanisms that is known to be revenue-optimal—we develop a non-clairvoyant dynamic mechanism that is robust to both estimation errors in the buyer's value distribution and strategic behavior on the part of the buyer. We then tailor its structure to achieve a policy with provably low regret against a constant approximation of the optimal dynamic mechanism in contextual auctions. Our result substantially improves on previous results that only provide revenue guarantees against static benchmarks.

## 1   Introduction

As a fundamental problem in mechanism design, pricing in repeated auctions has been extensively studied in recent years. This is partly motivated by the popularity of selling online ads via auctions, an industry totalling annual revenue of hundreds of billions of dollars. Repeated auctions open up the possibility of linking auctions across time using state information in order to enhance revenue or welfare, but this introduces several challenges. To guarantee optimal outcomes, the process must take into account the bidders' incentives to possibly manipulate each individual auction as well as the auction state across time. In practice, the seller must also rely on approximate models of the buyers' preferences to effectively set auction parameters like reserve prices. These aspects of the problem have so far been explored in two separate strands of the literature on repeated auctions, where items arrive online and the allocation and payment decisions must be made as soon as an item arrives.

One strand, known as *dynamic mechanism design*, considers an environment in which the seller has exact distributional information over the buyers' values for the items, for the current stage and all future stages, and designs revenue-maximizing dynamic mechanisms that adapt the auction state based on the buyer's historical bids [Thomas and Worrall, 1990, Bergemann and Välimäki, 2010, Ashlagi et al., 2016, Mirrokni et al., 2016a,b]. However, this *clairvoyant* framework relies on the seller having an accurate forecast of the buyer's valuation distributions in future auctions. To address this concern, Mirrokni et al. [2018] propose *non-clairvoyant* dynamic mechanisms, which do not rely on any information about the future (but do rely on an accurate forecast of the present). They show that a non-clairvoyant dynamic mechanism can achieve a constant approximation to the revenue of the optimal clairvoyant mechanism. The other strand of literature, known as *robust price learning*, focuses on a setting where the buyer's value distributions across stages are parameterized by some common private factors that are unknown to the seller, and designs robust policies to learn from the

buyer's bids and set prices with good revenue performance [Amin et al., 2013, 2014, Medina and Mohri, 2014, Golrezaei et al., 2018]. Although these results also take into account strategic buyer behavior, they only provide guarantees against the revenue-optimal *static* benchmark, which does not take advantage of auction state across time and whose revenue can be arbitrarily smaller than the optimal dynamic benchmark [Papadimitriou et al., 2016].

In this work, we consider a scenario in which the designer can only make use of an estimate of the buyer's value distribution in the present auction stage, which connects dynamic mechanism design with the problem of learning. Designing dynamic auctions in this setting is challenging for several reasons. When the seller's estimate of the distribution is not perfectly aligned with the buyer's true distribution, it is impossible for the seller to offer a dynamic mechanism that is exactly incentive-compatible and also makes use of the prior on values. Furthermore, unlike static mechanisms in which the auction for each item is independent of the buyer's past reports, in a dynamic mechanism a buyer's misreport can potentially affect auctions for all future items. We overcome these obstacles and provide a *robust non-clairvoyant* dynamic mechanism such that the extent of the buyers' misreports and the revenue loss can be related to and bounded by the estimation error. We then apply our robust dynamic mechanism to the concrete problem of contextual auctions, where a buyer's valuation depends on the context that describes the item, but the relationship between the buyer's valuation and the context is unknown to the seller and must be estimated across auctions. The seller's task is to design a policy which adapts the auction mechanism based on the buyer's historical bids, with the objective of maximizing revenue. Previous results give no-regret policies against the optimal *static* mechanism [Amin et al., 2014, Golrezaei et al., 2018], but as mentioned it is known that the revenue gap between optimal static and dynamic mechanisms can be arbitrarily large [Papadimitriou et al., 2016]. We tailor the structure of our robust non-clairvoyant dynamic mechanism to a learning environment, leading to a no-regret policy against the strong benchmark of a *constant approximation* of the *optimal clairvoyant dynamic mechanism*.

**Related Work**   We briefly discuss research in dynamic mechanism design that is closely related to the present work. For a comprehensive review of the literature readers are encouraged to refer to [Bergemann and Said, 2011]. Papadimitriou et al. [2016] provide an example demonstrating that the revenue gap between optimal static and the dynamic mechanisms can be arbitrarily large, which is a key motivation for the use of dynamic mechanisms in our setting. Moreover, they show that it is NP-Hard to design the optimal deterministic auctions even in an environment with a single buyer and two items only. Ashlagi et al. [2016] and Mirrokni et al. [2016b] simultaneously and independently provide a fully polynomial-time approximation scheme to compute the optimal randomized mechanism. Our work builds upon the framework of bank account mechanisms developed by Mirrokni et al. [2016a,b, 2018]. Based on the bank account mechanism, Mirrokni et al. [2018] design a non-clairvoyant mechanism achieving $1/3$ of the revenue of a clairvoyant mechanism. However, their mechanism relies on exact distributional information, which makes it unsuitable in a learning environment where value distributions are estimated. Our robust dynamic mechanism addresses this limitation.

Our work is closely related to dynamic pricing with learning; see [den Boer, 2015] for a recent survey. There has been a growing body of literature on learning in dynamic pricing in contextual auctions with non-strategic buyers [Cohen et al., 2016, Lobel et al., 2018, Leme and Schneider, 2018, Mao et al., 2018]. In their models, the buyers have homogeneous valuations and are non-strategic, and thus, the problem can be reduced to a single-item setting where the buyer acts myopically without considering the impact on the future auction from their current bids. However, Edelman and Ostrovsky [2007] provide empirical evidence that the buyers participating in the online advertising markets do act strategically. The study of robust price learning with strategic buyers was initiated by Amin et al. [2013] and Medina and Mohri [2014]. They design no-regret policies in a non-contextual environment where the buyer's valuation is fixed and the seller repeatedly interacts with a single buyer through posted price auctions, where the buyer is less patient than the seller. The regret guarantee is later improved to $\Theta(\log \log T)$ by Drutsa [2017, 2018]. Amin et al. [2013] show that no learning algorithm can achieve sublinear revenue loss if the buyer is as patient as the seller.

For learning in contextual auctions, Amin et al. [2014] develop a no-regret policy in a setting without market noise. Golrezaei et al. [2018] enrich the model by incorporating market noise and design a no-regret policy for cases where the market noise is known exactly or adversarially selected from a set of distributions. Liu et al. [2018] apply techniques from differential privacy to learn optimal

reserve prices against non-myopic bidders. All these results are no-regret against the optimal *static* mechanism as a benchmark, whereas our policy is no-regret against a constant-factor approximation of the optimal *dynamic* mechanism which has all distributional information available in advance.

## 2   Preliminaries

In a *dynamic* auction a seller (he) sells a stream of $T$ items that arrive online, based on bids placed by strategic buyers. An item must be sold when it arrives. For the sake of simplicity we will focus on the case of a single buyer (she) throughout this paper.[1] At the beginning of stage $t$ a new item arrives and the buyer's valuation $v_t \in [0, a_t]$ for the item is drawn independently from a distribution $F_t$ with density $f_t$. The distributions are not necessarily identical across stages. We assume that $f_t$ is continuous and upper bounded by $c_f/a_t$ where $c_f$ is a constant. The domain bounds $a_t$ are known to the seller and may vary across stages to reflect the fact that item valuations may have different scales.[2] As a special case of this framework, in a *contextual* auction the item at stage $t$ is represented by an observable feature vector $\boldsymbol{\zeta}_t \in \mathbb{R}^d$ with $\|\boldsymbol{\zeta}_t\|_2 \leq 1$. In line with the literature, we assume that the feature vectors are drawn independently from a fixed distribution $\mathcal{D}$ with positive-definite covariance matrix [Golrezaei et al., 2018]. The buyer's preferences are encoded by a fixed vector $\boldsymbol{\sigma} \in \mathbb{R}^d$ and the buyer's valuation at stage $t$ takes the form $v_t = a_t(\langle \boldsymbol{\sigma}, \boldsymbol{\zeta}_t \rangle + \varepsilon_t)$, where $\varepsilon_t$ is a noise term with cumulative distribution $M_t$. The distribution $M_t$ and the feature vector $\boldsymbol{\zeta}_t$ are observed by the seller but the buyer's preference vector $\boldsymbol{\sigma}$ remains private. We make the following technical assumption on the sequence of $a_t$:

**Assumption 1.** *For all $t$, $\sum_{t' \leq t} a_{t'} \leq c_a \cdot t$ where $c_a$ is a constant.*

Assumption 1 limits the portion of welfare and revenue that can arise in the first $t$ stages, for any $t$. Its purpose is to rule out situations where a large fraction of revenue comes from the initial stages, under which a large revenue loss may be inevitable since it is impossible for the seller to obtain a good estimate of $\boldsymbol{\sigma}$ from just the first few stages.

Once the buyer learns her valuation $v_t$ at stage $t$, she then submits a bid $b_t \in [0, a_t]$ to the seller who then decides whether to allocate the item (perhaps stochastically) and what payment to charge. We write $V^t$ to denote the set of all possible sequences $(b_1, \ldots, b_t)$ of buyer bids for the first $t$ stages, and similarly we write $(\Delta V)^t$ to denote the set of all possible independent distributions over the sequence of first $t$ bids. The seller's distributional beliefs over the buyer's values across stages are denoted as $\hat{F}_{(1,T)} = (\hat{F}_1, \hat{F}_2, \ldots, \hat{F}_T)$. Throughout the paper we will use the notation $\hat{F}_{(t',t'')}$ to represent $(\hat{F}_{t'}, \ldots, \hat{F}_{t''})$, and similarly for $F_{(t',t'')}$, $v_{(t',t'')}$, and $b_{(t',t'')}$. A dynamic mechanism is represented by sequences $(x_1, \ldots, x_T)$ and $(p_1, \ldots, p_T)$ where $x_t$ and $p_t$ denote the allocation rule and the payment rule at stage $t$, respectively. We refer to $\langle x_t, p_t \rangle$ as the *stage mechanism* at stage $t$.

**Non-Clairvoyant Dynamic Mechanism.** In a non-clairvoyant environment, the seller obtains an estimated distribution $\hat{F}_t$ only at stage $t$ and not before, so the mechanism at stage $t$ can only depend on $\hat{F}_{(1,t)}$. The allocation function $x_t$ maps the history of bids $b_{(1,t)}$ and distribution $\hat{F}_{(1,t)}$ to an allocation probability, $x_t : V^t \times (\Delta V)^t \to [0, 1]$. The payment function $p_t$ maps the history of bids $b_{(1,t)}$ and the distribution $\hat{F}_{(1,t)}$ to a real-valued payment, $p_t : V^t \times (\Delta V)^t \to \mathbb{R}$. In line with the literature, we assume the buyer has a quasi-linear utility such that the buyer's utility from bidding $b_t$ at stage $t$ is $u_t\big(v_t; b_{(1,t)}; \hat{F}_{(1,t)}\big) = v_t \cdot x_t\big(b_{(1,t)}; \hat{F}_{(1,t)}\big) - p_t\big(b_{(1,t)}; \hat{F}_{(1,t)}\big)$. In the contextual auction setting the seller maintains a model $\hat{\boldsymbol{\sigma}}_t$ for the buyer's preference vector estimated from prior bidding behavior, and combines with $a_t$, $\boldsymbol{\zeta}_t$, and noise model $M_t$, which can only be observed at the beginning of stage $t$ and not before, to compute $\hat{F}_t$.

**Utility-Maximizing Buyer.** We assume that the buyer knows the true distributions $F_{(1,T)}$ in advance so that she can reason about how the mechanism will evolve over time and compute a bidding strategy that maximizes her utility. Specifically, we consider a buyer who aims to maximize her time discounted utility $\sum_{t'=t}^T \gamma^{t'-t} \cdot \mathbb{E}[u_t]$ at stage $t$ where $\gamma \in [0, 1)$ is the discounting factor and the

expectation is taken with respect to $F_{(1,T)}$. We note that it is impossible to obtain a no-regret policy when the buyer is as patient as the seller (the case of $\gamma = 1$) [Amin et al., 2013].

**Incentive Constraints.** In a dynamic environment, the buyer's best response at stage $t$ depends on her strategy in the future stages. When the seller has perfect distributional information, the classic notion of dynamic incentive-compatibility (DIC) requires that the buyer is incentivized to report truthfully assuming that she plays optimally in the future [Mirrokni et al., 2018].[3] When the seller only has approximate distributional information this is no longer possible to achieve, so we introduce the notion of $\eta_{(1,T)}$-approximate DIC, which requires that the buyer's bid deviate from the truth by at most $\eta_t$ at stage $t$, assuming the buyer plays optimally in the future (note that optimally now no longer means truthfully). Formally, at each stage $t$, there exists $\hat{b}_t \in [v_t - \eta_t, v_t + \eta_t]$ such that

$$\hat{b}_t \in \arg\max_{b_t} u_t\big(v_t; b_{(1,t)}; \hat{F}_{(1,t)}\big) + \gamma \cdot U_t\big(b_{(1,t)}; F_{(1,T)}; \hat{F}_{(1,T)}\big) \qquad (\eta_{(1,T)}\text{-DIC})$$

for all $v_t$, $b_{(1,t-1)}$, $F_{(t+1,T)}$, and $\hat{F}_{(t+1,T)}$, where $U_t(b_{(1,t)}; F_{(1,T)}; \hat{F}_{(1,T)})$ is the continuation utility that the buyer obtains in the future: $U_T\big(b_{(1,T)}; F_{(1,T)}; \hat{F}_{(1,T)}\big) = 0$, and for $t < T$ $U_t\big(b_{(1,t)}; F_{(1,T)}; \hat{F}_{(1,T)}\big)$ is defined as

$$\mathbb{E}_{v_{t+1} \sim F_{t+1}} \Big[ \max_{b_{t+1}} u_{t+1}\big(v_{t+1}; b_{(1,t+1)}; \hat{F}_{(1,t+1)}\big) + \gamma \cdot U_{t+1}\big(b_{(1,t+1)}; F_{(1,T)}; \hat{F}_{(1,T)}\big) \Big].$$

**Participation Constraints.** We assume that the buyer weighs realized past utilities equally. Therefore, ex-post individual rationality requires that for all $\hat{F}_{(1,T)}$ and for all $v_{(1,T)}$,

$$\sum_{t=1}^{T} u_t\big(v_t; v_{(1,t)}; \hat{F}_{(1,t)}\big) \geq 0. \qquad \text{(ex-post IR)}$$

For convenience, we will use the phrase "for $F_{(1,T)}$" to indicate the environment where the buyer's true distribution is $F_{(1,T)}$. For example, when we say that a mechanism is $\eta_{(1,T)}$-DIC for $F_{(1,T)}$ we mean that it is $\eta_{(1,T)}$-DIC when the buyer's true distribution is $F_{(1,T)}$.

**No-Regret Policy.** Our task is to design a policy $\pi$ that includes both a learning policy for $\boldsymbol{\sigma}$ and an associated dynamic mechanism policy to extract revenue. At the beginning of stage $t$, the learning policy estimates $\hat{F}_t$ using information $a_{(1,t)}$, $\boldsymbol{\zeta}_{(1,t)}$, $M_{(1,t)}$, and $b_{(1,t-1)}$, while the dynamic mechanism policy computes the stage mechanism $\langle x_t, p_t \rangle$ at stage $t$ using $\hat{F}_{(1,t)}$ and $b_{(1,t-1)}$. Let $\text{Rev}(\pi; F_{(1,T)})$ and $\text{Rev}(B; F_{(1,T)})$ be the revenue of implementing policy $\pi$ and mechanism $B$ for $F_{(1,T)}$, respectively. Moreover, let $B^*(F_{(1,T)})$ denote the revenue-optimal *clairvoyant* dynamic mechanism that knows $F_{(1,T)}$ in advance. The regret of policy $\pi$ against a $c$-approximation of the dynamic benchmark is defined as $\text{Regret}^{\pi}(F_{(1,T)}) = c \cdot \text{Rev}\big(B^*(F_{(1,T)}); F_{(1,T)}\big) - \text{Rev}(\pi; F_{(1,T)})$. Our objective is to design a policy with sublinear regret.[4]

## 3 Robust Non-clairvoyant Mechanism

The literature on dynamic mechanism design relies on the strong assumption that the seller has perfect distributional information at each stage, $\hat{F}_{(1,T)} = F_{(1,T)}$ [Ashlagi et al., 2016, Mirrokni et al., 2016b,a, 2018]. However, in a learning setting like that of contextual auctions, the seller can only obtain a sequence of estimated distributions by estimating $\boldsymbol{\sigma}$. In this section, we design a non-clairvoyant mechanism that is robust to misspecifications in the value distribution in the sense that the buyer is incentivized to place a bid within known bounds from its value, which ultimately allows us to relate the mechanism revenue under the estimated and true value distributions. The misspecifications handled by the mechanism are captured by the following assumption.

**Assumption 2.** *There exists a coupling between a random draw $v_t$ from $F_t$ and a random draw $\hat{v}_t$ from $\hat{F}_t$ such that $v_t = \hat{v}_t + a_t \cdot \epsilon_t$ with $\epsilon_t \in [-\Delta, \Delta]$.*

### 3.1 The Mechanism

Building on the $\frac{1}{3}$-approximation non-clairvoyant mechanism from Mirrokni et al. [2018], we design our robust non-clairvoyant mechanism by mixing their mechanism with a random posted-price auction. The mechanism is an instance of a *bank account mechanism* where the state information is captured by a single scalar $\mathrm{bal}_t$.

**Mechanism 1.** *The robust non-clairvoyant mechanism $B(\hat{F}_{(1,T)}, \lambda)$ consists of a mixture of four mechanisms: the give-for-free mechanism, the posted-price auction with extra fee, the Myerson's auction, and the random posted-price auction. The stage mechanism at stage $t$ is parameterized by a non-negative balance $\mathrm{bal}_t$. When the buyer submits a bid $b_t$:*

*Give-for-free Mechanism. Allocate the item no matter what the buyer's bid is and increase the balance by the buyer's bid: $x_t^G = 1$, $p_t^G = 0$, and $\mathrm{bal}_{t+1}^G = \mathrm{bal}_t + b_t$*

*Posted-price Auction with Extra Fee. Let $\mathrm{fee}_t(\mathrm{bal}_t; \hat{F}_t) = \min(3\mathrm{bal}_t, \mathbb{E}_{v_t \sim \hat{F}_t}[v_t])$ and $r_t(\mathrm{bal}_t)$ be the posted-price such that $\mathbb{E}_{v_t \sim \hat{F}_t}\left[\left(v_t - r_t(\mathrm{bal}_t)\right)^+\right] = \mathrm{fee}_t(\mathrm{bal}_t; \hat{F}_t)$. The mechanism charges the buyer $\mathrm{fee}_t(\mathrm{bal}_t; \hat{F}_t)$ before the buyer learns her valuation and then runs a posted-price auction with price $r_t(\mathrm{bal}_t)$: $x_t^P = \mathbf{1}\{b_t \geq r_t(\mathrm{bal}_t)\}$ and $p_t^P = \mathrm{fee}_t(\mathrm{bal}_t; \hat{F}_t) + r_t(\mathrm{bal}_t) \cdot \mathbf{1}\{b_t \geq r_t(\mathrm{bal}_t)\}$, and decrease the balance by $\mathrm{fee}_t(\mathrm{bal}_t; \hat{F}_t)$: $\mathrm{bal}_{t+1}^P = \mathrm{bal}_t - \mathrm{fee}_t(\mathrm{bal}_t; \hat{F}_t)$.*

*Myerson's Auction. Let $r_t^*(\hat{F}_t)$ be Myerson's optimal reserve price, i.e., $r_t^*(\hat{F}_t) = \arg\max_r r \cdot \left(1 - \hat{F}_t(r)\right)$ and run a posted-price auction with price $r_t^*(\hat{F}_t)$ without changing the balance: $x_t^M = \mathbf{1}\{b_t \geq r_t^*(\hat{F}_t)\}$, $p_t^M = r_t^*(\hat{F}_t) \cdot \mathbf{1}\{b_t \geq r_t^*(\hat{F}_t)\}$, and $\mathrm{bal}_{t+1}^M = \mathrm{bal}_t$.*

*Random Posted-price Auction. Let $\hat{r}_t$ be random reserve price drawn from $[0, a_t]$ uniformly and run a posted-price auction with price $\hat{r}_t$ without changing the balance: $x_t^R = \mathbf{1}\{b_t \geq \hat{r}_t\}$, $p_t^R = \hat{r}_t \cdot \mathbf{1}\{b_t \geq \hat{r}_t\}$, and $\mathrm{bal}_{t+1}^R = \mathrm{bal}_t$.*

*The robust non-clairvoyant mechanism at stage $t$ is: $x_t = \lambda \cdot x_t^R + \frac{1-\lambda}{3}\left[x_t^G + x_t^P + x_t^M\right]$, $p_t = \lambda \cdot p_t^R + \frac{1-\lambda}{3}\left[p_t^G + p_t^P + p_t^M\right]$, and $\mathrm{bal}_t = \lambda \cdot \mathrm{bal}_t^R + \frac{1-\lambda}{3}\left[\mathrm{bal}_t^G + \mathrm{bal}_t^P + \mathrm{bal}_t^M\right]$.*

The following central result gives a guarantee on the revenue performance of our robust non-clairvoyant mechanism against a utility-maximizing buyer subject to an estimation error $\Delta$.

**Theorem 3.1.** $\mathrm{Rev}\left(B(\hat{F}_{(1,T)}, \lambda), F_{(1,T)}\right) \geq \frac{1}{3}\mathrm{Rev}\left(B^*(F_{(1,T)}), F_{(1,T)}\right) - O\left(\lambda T + \sqrt{\frac{\Delta}{\lambda}}T\right)$.

At the optimal choice of $\lambda = \Delta^{\frac{1}{3}}$ the revenue loss is $O\left(\Delta^{\frac{1}{3}}T\right)$. The remainder of this section is devoted to proving Theorem 3.1.

### 3.2 Analysis

We start by describing the incentive properties that $B(\hat{F}_{(1,T)}, \lambda)$ satisfies for $\hat{F}_{(1,T)}$. First notice that all four base mechanisms are variants of posted-price auctions, and therefore, all of them are stage-IC:

$$\forall b_t, \ v_t \cdot x_t(\mathrm{bal}, v_t) - p_t(\mathrm{bal}, v_t) \geq v_t \cdot x_t(\mathrm{bal}, b_t) - p_t(\mathrm{bal}, b_t). \quad \text{(stage-IC)}$$

In particular, all mechanisms except the posted-price auction with extra fee are stage-IR:

$$\forall v_t, \ v_t \cdot x_t(v_t) - p_t(v_t) \geq 0 \quad \text{(stage-IR)}$$

We emphasize that the posted-price auction with extra fee is different from a classic posted-price auction: the posted-price auction with extra fee will charge the buyer an extra payment $\mathrm{fee}_t(\mathrm{bal}_t; \hat{F}_t)$ no matter what the buyer's bid is, and therefore, it is not stage-IR. Moreover, each stage mechanism is balance-independent (BI) with respect to the estimated distribution $\hat{F}_t$: there exists a constant $c_t$,

$$\mathbb{E}_{v_t \sim \hat{F}_t}[v_t \cdot x_t(\mathrm{bal}, v_t) - p_t(\mathrm{bal}, v_t)] = c_t. \quad \text{(BI)}$$

In particular, the give-for-free mechanism, the Myerson's auction, and the random posted-price auction are static and independent of the balance; as for the posted-price auction with extra fee, it ensures that the buyer's expected utility is always $0$ for all $\mathrm{bal}_t \geq 0$ under $\hat{F}_t$.

The combination of stage-IC and BI implies that the mechanism is DIC: since the mechanism promises the buyer that all future stage mechanisms are BI, the buyer can infer that her action at the current stage does not impact her expected utility in the future. Moreover, notice that the non-negative balance $\mathrm{bal}$ always lower-bounds the buyer's cumulative utility, and therefore, $B(\hat{F}_{(1,T)}, \lambda)$ is ex-post IR under the estimated distributions $\hat{F}_{(1,T)}$.

**Proposition 3.1.** $B(\hat{F}_{(1,T)}, \lambda)$ *is stage-IC, BI, DIC, and ex-post IR for* $\hat{F}_{(1,T)}$.

We next turn to the mechanism's properties under the true distributions $F_{(1,T)}$.

### 3.2.1 Mismatch between $\hat{F}_{(1,T)}$ and $F_{(1,T)}$

We first bound the revenue loss due to the mismatch between $\hat{F}_{(1,T)}$ and $F_{(1,T)}$. Observe that one can interpret the estimation error under Assumption 2 as the buyer's misreport: when the buyer reports truthfully under $F_{(1,T)}$ this is equivalent to the case in which the buyer misreports by a magnitude at most $a_t \cdot \Delta$ under $\hat{F}_{(1,T)}$. We develop a program for computing the revenue of our mechanism even when the buyer misreports. For a non-clairvoyant mechanism $B(\hat{F}_{(1,T)}, \lambda)$, we consider a program $\psi_t(\mathrm{bal}, \hat{F}_{(1,T)}; F_{(1,T)})$ to keep track on the revenue of implementing $B(\hat{F}_{(1,T)}, \lambda)$ when the buyer's true distribution is $F_{(1,T)}$. We define $\psi_T(\mathrm{bal}) = 0$ and for $t < T$,

$$\psi_{t-1}(\mathrm{bal}, \hat{F}_{(1,T)}; F_{(1,T)}) = \mathbb{E}_{v_t \sim F_t}\left[\frac{1}{3}\mathsf{fee}_t(\mathrm{bal}; \hat{F}_t) + \frac{1}{3}r_t^*(\hat{F}_t) \cdot \mathbf{1}\{v_t' \geq r_t^*(\hat{F}_t)\} \right.$$
$$\left. + \psi_t\left(\mathrm{bal} + \frac{1}{3}v_t' - \frac{1}{3}\mathsf{fee}_t(\mathrm{bal}; \hat{F}_t), \hat{F}_{(1,T)}; F_{(1,T)}\right)\right] \quad (1)$$

where $v_t'$ is the buyer's reported bid that maximizes her continuation utility when her true value is $v_t$.

Recall that conditioned on that the stage mechanism is not the random posted-price auction, with $\frac{1}{3}$ probability, we run the posted-price auction with extra fee and extract $\mathsf{fee}_t(\mathrm{bal}; \hat{F}_t)$ as revenue. Here, we omit the revenue $r_t(\mathrm{bal}_t)$ obtained from the posted-price auction with extra fee. In addition, with another $\frac{1}{3}$ probability, we run the Myerson's auction and extract $r_t^*(\hat{F}_t)$ revenue if $v_t' \geq r_t^*(\hat{F}_t)$. Moreover, the balance is increased by $v_t'$ with probability $\frac{1}{3}$ from the give-for-free mechanism and decreased by $\frac{1}{3}\mathsf{fee}_t(\mathrm{bal})$ with probability $\frac{1}{3}$ from the posted-price auction.

**Proposition 3.2.** $\mathrm{Rev}\left(B(\hat{F}_{(1,T)}, \lambda); F_{(1,T)}\right) \geq (1 - \lambda) \cdot \psi_0(0, \hat{F}_{(1,T)}; F_{(1,T)})$.

According to the revenue analysis in [Mirrokni et al., 2018], we can still obtain $\frac{1}{3}$-approximation of the optimal revenue even when the revenue $r_t(\mathrm{bal}_t)$ is omitted.

**Lemma 3.1.** *[Mirrokni et al., 2018]* $\psi_0(0, F_{(1,T)}; F_{(1,T)}) \geq \frac{1}{3} \cdot \mathrm{Rev}\left(B^*(F_{(1,T)}), F_{(1,T)}\right)$.

The following lemma establishes a connection between the change of the balance and the change of the revenue, when the seller's distributional information is perfect so that the buyer does not misreport. In particular, it shows that as balance increases by $\delta$, the change of the future revenue is between $0$ and $\delta$. Therefore, it demonstrates the smoothness of revenue curve such that if the buyer misreports at stage $t$ to change the balance by $\delta$, then the revenue loss is at most $\delta$ for the future stages, assuming the buyer reports truthfully in the future.

**Lemma 3.2.** *For all* $0 \leq t \leq T$ *and* $\delta \geq 0$,

$$\psi_t(\mathrm{bal} + \delta, F_{(1,T)}; F_{(1,T)}) - \delta \leq \psi_t(\mathrm{bal}, F_{(1,T)}; F_{(1,T)}) \leq \psi_t(\mathrm{bal} + \delta, F_{(1,T)}; F_{(1,T)}).$$

Applying Lemma 3.2 with Assumption 2, we can bound the revenue loss due to the mismatch between $F_{(1,T)}$ and $\hat{F}_{(1,T)}$. More precisely, we will bound the difference between $\psi_0(0, F_{(1,T)}; F_{(1,T)})$ and $\psi_0(0, \hat{F}_{(1,T)}; \hat{F}_{(1,T)})$. Notice that $B(F_{(1,T)})$ is dynamic incentive-compatible for $F_{(1,T)}$, and thus, the buyer will not misreport, i.e., $v_t' = v_t$ in (1); similarly for $\hat{F}_{(1,T)}$.

**Lemma 3.3.** $\psi_0(0, \hat{F}_{(1,T)}; \hat{F}_{(1,T)}) \geq \psi_0(0, F_{(1,T)}; F_{(1,T)}) - O(\Delta T)$.

### 3.2.2 The Buyer's Misreport

Note that in a single-buyer environment, the properties stage-IC and ex-post IR do not depend on the underlying distributions, and therefore, $B(\hat{F}_{(1,T)}, \lambda)$ is also stage-IC and ex-post IR for $F_{(1,T)}$. However, $B(\hat{F}_{(1,T)}, \lambda)$ is no longer BI for $F_{(1,T)}$, which is the key property to ensure DIC. To circumvent this difficulty, we generalize the definition of BI to *approximate balance-independence*.

**Definition 3.1.** *A dynamic mechanism is* $\beta_{(1,T)}$*-BI for* $F_{(1,T)}$ *if* $\forall t$*, there exists a constant* $c_t$*:*

$$\forall \text{bal} \geq 0, \mathbb{E}_{v_t \sim F_t}[v_t \cdot x_t(\text{bal}, v_t) - p_t(\text{bal}, v_t)] \in [c_t - \frac{\beta_t}{2}, c_t + \frac{\beta_t}{2}] \qquad (\beta_{(1,T)}\text{-BI})$$

Since with the same stage mechanism, the difference between the expected utility under $\hat{F}_t$ and $F_t$ is at most $\Delta a_t$, $B(\hat{F}_{(1,T)}, \lambda)$ is $\beta_{(1,T)}$-BI with $\beta_t = 2\Delta a_t$.

**Proposition 3.3.** $B(\hat{F}_{(1,T)}, \lambda)$ *is stage-IC,* $\beta_{(1,T)}$*-BI with* $\beta_t = 2\Delta a_t$*, and ex-post IR for* $F_{(1,T)}$*.*

For a dynamic mechanism satisfying $\beta_{(1,T)}$-BI for $F_{(1,T)}$, the range of the buyer's expected utility under truthful reporting is $\beta_t$ in the $t$-th stage. Therefore, no matter how the buyer misreports in the first $(t-1)$ stages, her expected utility in the $t$-th stage can only fluctuate at most $\beta_t$ if she reports truthfully at stage $t$. Combining this with the fact that the stage mechanisms are stage-IC, we have

**Lemma 3.4.** *For a dynamic mechanism that is stage-IC and* $\beta_{(1,T)}$*-BI for* $F_{(1,T)}$*, for any* $b_{(1,t-1)}$ *and* $v_t$*, the difference between the continuation utility of reporting any* $b_t \in [0, a_t]$ *and the continuation utility of reporting* $v_t$ *truthfully is bounded by* $\sum_{t'=t+1}^{T} \gamma^{t'-t} \cdot \beta_{t'}$*.*

Lemma 3.4 states that the gain of the continuation utility by misreporting is bounded and the bound is independent of the magnitude of the misreport. The key observation behind Lemma 3.4 is that at stage $t$, the buyer obtains the maximum utility when she reports truthfully since the stage mechanism is stage-IC. Therefore, by the property of $\beta_{(1,T)}$-BI, the difference of utility between misreporting in an optimal way and reporting truthfully is at most $\beta_t$ at stage $t$.

As a result, once the mechanism posts a risk for misreporting, we are able to bound the magnitude of the buyer's misreport. This is the purpose of mixing in the random posted-price mechanism at each stage $t$: it can be shown that a misreport with magnitude $m_t$ will cause the buyer a utility loss $\lambda \cdot \frac{m_t^2}{2a_t}$. Since the buyer is a utility-maximizer with discounting factor $\gamma$, we can bound the magnitude of misreport for each stage:

**Lemma 3.5.** $B(\hat{F}_{(1,T)}, \lambda)$ *is* $\eta_{(1,T)}$*-DIC with* $\eta_t = \sqrt{\frac{2a_t}{\lambda} \cdot \sum_{t'=t+1}^{T} \gamma^{t'-t} \beta_{t'}}$*.*

Applying Lemma 3.2, we can show that $B(\hat{F}_{(1,T)}, \lambda)$ is robust against the buyer's misreport. We abuse the notion to use $\psi_0(0, \hat{F}_{(1,T)}; F_{(1,T)})$ to track the revenue conditioned on that the magnitude of the buyer's misreport at stage $t$ is bounded by $\eta_t$.

**Lemma 3.6.** $\psi_0(0, \hat{F}_{(1,T)}; F_{(1,T)}) \geq \psi_0(0, \hat{F}_{(1,T)}; \hat{F}_{(1,T)}) - O\left(\sqrt{\frac{\Delta}{\lambda}} T\right)$*.*

Finally, combining Proposition 3.2, Lemma 3.3 and Lemma 3.6, completes the proof of Theorem 3.1.

## 4 No-Regret Policy in Contextual Auctions

### 4.1 Learning Policy

Our learning policy is adapted from the contextual robust pricing policy proposed in [Golrezaei et al., 2018]. Our learning policy partitions the entire time horizon into $K = \lceil \log T \rceil$ phases where $T$ is the time horizon, such that the partition is specified by $(\ell_1 = 1, \ell_2, \cdots, \ell_K, \ell_{K+1} = T + 1)$, in which $\ell_k = 2^{k-1}$. The $k$-th phase spans between the $\ell_k$-th stage and the $(\ell_{k+1} - 1)$-th stage, and therefore, the length of phase $k$ is exactly $\ell_k$. Note that the partition can be implemented even when $T$ is not known in advance. We use $E_k = \{\ell_k, \cdots, \ell_{k+1} - 1\}$ to refer to the stages in the $k$-th phase.

At the beginning of the $k$-th phase, we update the estimation of the buyer's preference vector $\boldsymbol{\sigma}$ using the buyer's bids from the $(k-1)$-th phase, denoted by $\hat{\boldsymbol{\sigma}}_k$. To estimate $\hat{\boldsymbol{\sigma}}_k$, we sample $w_t$ uniformly

from $[0, 1]$ for $t \in \hat{E}_{k-1}$, where $\hat{E}_{k-1} = \{t \in E_{k-1} \mid \ell_k - t > c \log \ell_k\}$ for some constant $c$. In other words, we will only use the information from the stages that are at least $c \log \ell_k$ ahead of the end of phase $(k-1)$. $\hat{\boldsymbol{\sigma}}_k$ is set to be $\arg \min_{\|\boldsymbol{\sigma}\| \leq 1} \mathcal{L}_{k-1}(\boldsymbol{\sigma})$, where

$$\mathcal{L}_{k-1}(\boldsymbol{\sigma}) = - \sum_{t \in \hat{E}_{k-1}} \Big[ \mathbf{1}\{b_t \geq a_t \cdot w_t\} \log\big(1 - M_t(w_t - \langle \boldsymbol{\sigma}, \boldsymbol{\zeta}_t \rangle)\big)$$

$$+ \mathbf{1}\{b_t < a_t \cdot w_t\} \log\big(M_t(w_t - \langle \boldsymbol{\sigma}, \boldsymbol{\zeta}_t \rangle)\big)\Big].$$

Note that when the buyer reports truthfully, $\mathcal{L}_{k-1}(\boldsymbol{\sigma})$ is exactly the negative of log-likelihood corresponding to $\boldsymbol{\sigma}$. We do not change our estimation throughout the $k$-th phase and the next update happens at the beginning of the $(k+1)$-phase. As a result, based on the estimate $\hat{\boldsymbol{\sigma}}_k$, we compute the estimated distribution in phase $k$ as $\hat{F}_t(v_t) = M_t\left(\frac{v_t}{a_t} - \langle \hat{\boldsymbol{\sigma}}_k, \boldsymbol{\zeta}_t \rangle\right)$ for all $t \in E_k$.

We say a *lie* is a misreport from the buyer that results in $\mathbf{1}\{b_t \geq a_t \cdot w_t\} \neq \mathbf{1}\{v_t \geq a_t \cdot w_t\}$. Let

$$L_{k-1} = \big\{t \in \hat{E}_{k-1} \mid \mathbf{1}\{b_t \geq a_t \cdot w_t\} \neq \mathbf{1}\{v_t \geq a_t \cdot w_t\}\big\}$$

be the set of stages in which the buyer lies. For a dynamic mechanism that is $\eta_{(1,T)}$-DIC, we have $v_t - \eta_t \leq b_t \leq v_t + \eta_t$. Hence, if $|a_t \cdot w_t - v_t| > \eta_t$, any misreport from the buyer does not result in a lie. Moreover, the buyer has an additional motivation to misreport to change the seller's estimation for the future phases. However, for $t \in \hat{E}_{k-1}$, such a gain is relatively small since the buyer discounts the future.

Let $B(\hat{F}_{(1,T)}, \lambda_{(1,K)})$ be a mechanism generalized from $B(\hat{F}_{(1,T)}, \lambda)$ such that for $t \in E_k$, $B(\hat{F}_{(1,T)}, \lambda_{(1,K)})$ offers the random posted-price auction with probability $\lambda_k$ instead of $\lambda$.

**Lemma 4.1.** *In $B(\hat{F}_{(1,T)}, \lambda_{(1,K)})$, the additional misreport at stage $t \in \hat{E}_k$ is $O(\frac{1}{\sqrt{\lambda_k} \cdot \ell_k^2})$. Moreover,* $|L_k| = O\left(\log \ell_k + \sum_{t \in \hat{E}_k} \frac{\eta_t}{a_t}\right)$ *with probability $1 - \frac{1}{\ell_k}$.*

Given this upper bound on $|L_{k-1}|$, the following lemma bounds the estimation error of $\hat{\boldsymbol{\sigma}}_k$.

**Lemma 4.2** (Proposition 7.1 [Golrezaei et al., 2018]). *With probability $1 - \frac{1}{\ell_k}$, the estimation error for phase $k$ is $\Delta_k \equiv \|\hat{\boldsymbol{\sigma}}_k - \boldsymbol{\sigma}\| = O\left(d \cdot \frac{|L_{k-1}|}{\ell_{k-1}} + \sqrt{\frac{\log(\ell_{k-1} \cdot d)}{\ell_{k-1}}}\right)$.*

## 4.2 Dynamic Mechanism Policy

We develop a hybrid non-clairvoyant mechanism to reduce the number of lies by reducing the magnitude of misreports. To do so, observe that the buyer has no incentive to misreport in order to affect future stage mechanisms when the latter are static. However, as previously mentioned, offering a purely static mechanism may forego a large amount of revenue [Papadimitriou et al., 2016]. Motivated by this insight, our hybrid mechanism contains both dynamic stages dependent on the history and static stages independent of the history. We adapt $B(\hat{F}_{(1,T)}, \lambda_{(1,K)})$ to obtain a hybrid non-clairvoyant mechanism $B^{hybrid}(\hat{F}_{(1,T)}, \lambda_{(1,K)}, \omega, \tau)$, which is parameterized by $\omega \in (0, 1)$ and a function $\tau : \mathbb{Z}_+ \to \mathbb{R}_+$ that maps the phase number to a real number. The stage mechanism at stage $t$ is parameterized by $a_t$, two balances $\text{bal}_t$ and $\text{sbal}_t$, and an additional parameter $\text{sw}_t$. We provide a high level description of our mechanism while a detailed description is deferred to the full version.

Let $E_k^\omega = \{t \in E_k \mid a_t < \ell_k^\omega\}$. Intuitively, the hybrid non-clairvoyant mechanism runs different stage mechanisms conditioned on whether $t \in E_k^\omega$ or not: the stage mechanism is dynamic for $t \notin E_k^\omega$ and the stage mechanism is static for $t \in E_k^\omega$ with high probability.

More precisely, for $t \notin E_k^\omega$, the stage mechanisms are exactly the same as $B(\hat{F}_{(1,T)}, \lambda_{(1,K)})$ and in particular, the posted-price auction with extra fee only uses the balance from $\text{bal}_t$. For $t \in E_k^\omega$, the give-for-free mechanism and the Myerson's auction remain the same. We use $\text{sw}_t$ to keep track of the summation of expected valuations, i.e., $\text{sw}_t = \frac{1}{3} \sum_{t' \in E_k^\omega, t' < t} \mathbb{E}_{v_{t'} \sim \hat{F}_{t'}}[v_{t'}]$. If $\text{sw}_t < \tau(k)$, we turn the posted-price auction with extra fee into a give-for-free mechanism, but we increase the balance $\text{sbal}$ instead of $\text{bal}$; otherwise, we run the posted-price auction with extra fee, except that it only uses the balance from $\text{sbal}$ and it will in addition deposit the buyer's utility to $\text{sbal}$.

For $t \in E_k^\omega$ and $\mathrm{sw}_t < \tau(k)$, the stage mechanism is static since it in fact runs a give-for-free mechanism with probability $\frac{2(1-\lambda_k)}{3}$ and a Myerson's auction with probability $\frac{1-\lambda_k}{3}$, both of which are independent of the history. For $t \in E_k^\omega$ and $\mathrm{sw}_t \geq \tau(k)$, by choosing $\tau$ properly, we show that with high probability, even if the buyer plays strategically, $3\mathrm{sbal}_t \geq \mathbb{E}_{v_t \sim \hat{F}_t}[v_t]$, which implies that $\min\left(3\mathrm{sbal}_t, \mathbb{E}_{v_t \sim \hat{F}_t}[v_t]\right) = \mathbb{E}_{v_t \sim \hat{F}_t}[v_t]$ so that the posted-price would be 0. Therefore, with high probability, the hybrid posted-price auction with extra fee is a give-for-free mechanism with fee $\mathbb{E}_{v_t \sim \hat{F}_t}[v_t]$, which is static and independent of $\mathrm{bal}_t$ and $\mathrm{sbal}_t$. To formally prove these statements, we exploit the fact that the dynamics of $\mathrm{sbal}_t$ forms a martingale for stage $t$ with $\mathrm{sw}_t \geq \tau(k)$.

**Lemma 4.3.** *With* $\tau(k) = \Omega\left(\ell_k^{\frac{1}{2}(1+\omega)}\sqrt{\log \ell_k} + \sqrt{\frac{\Delta_k}{\lambda_k}}\ell_k\right)$ *for all* $k$, *we have*

$$\mathsf{Rev}\left(B^{hybrid}(\hat{F}_{(1,T)}, \lambda_{(1,K)}, \omega, \tau), F_{(1,T)}\right) \geq \frac{1}{3}\mathsf{Rev}\left(B^*(F_{(1,T)}), F_{(1,T)}\right) - \sum_k \left(\tau(k) + \lambda_k \cdot \ell_k\right)$$

*and with probability at least* $1 - \frac{1}{\ell_k}$, $\sum_{t \in \hat{E}_k} \frac{\eta_t}{a_t} \leq \tilde{O}\left(\ell_k^{1-\omega}\right)$.

Lemma 4.3 states that there exits a function $\gamma$ such that the revenue loss is at most $\sum_k \left(\tau(k) + \lambda_k \cdot \ell_k\right)$ and the number of lies is $\tilde{O}(\ell_k^{1-\omega})$. In particular, as $\omega$ increases, the revenue loss increases while the number of lies decreases, and therefore, our hybrid non-clairvoyant mechanism achieves a trade-off between the revenue loss and the number of lies.

### 4.3 The Final Policy

---

*Learning Policy*: At the start of phase $k$, estimate $\hat{\boldsymbol{\sigma}}_k = \arg\min_{\|\boldsymbol{\sigma}\| \leq 1} \mathcal{L}_{k-1}(\boldsymbol{\sigma})$.

*Dynamic Mechanism Policy*: $B^{hybrid}(\hat{F}_{(1,T)}, \lambda_{(1,K)}, \frac{1}{2}, \tau)$: at phase $k$

- $\lambda_k = \ell_k^{-\frac{1}{6}}$ and $\tau(k) = c^* \ell_k^{\frac{5}{6}}$;
- Compute the distributional information $\hat{F}_t$ for $t \in E_k$ according to the estimation $\hat{\boldsymbol{\sigma}}_k$;

---

Figure 1: Robust Non-clairvoyant Dynamic Contextual Auction Policy

We are now ready to combine our learning policy and dynamic mechanism policy to obtain our no-regret policy for contextual auctions in a non-clairvoyant environment (Figure 1). For our hybrid non-clairvoyant mechanism, we will set $\omega = \frac{1}{2}$, $\lambda_k = \ell_k^{-\frac{1}{6}}$, and $\tau(k) = c^* \ell_k^{\frac{5}{6}}$ with a large enough constant $c^*$. In particular, the estimation error for $\hat{\boldsymbol{\sigma}}_k$ is $\Delta_k = O(\ell_k^{-\frac{1}{2}})$ under our policy.

**Theorem 4.1.** *The $T$-stage regret of the robust non-clairvoyant dynamic contextual auction policy is* $\tilde{O}(T^{\frac{5}{6}})$ *against* $\frac{1}{3}$-*approximation of the optimal clairvoyant dynamic mechanism.*

## 5 Conclusion

In this paper, we present a framework of designing non-clairvoyant dynamic mechanisms that are robust to both the estimation errors on the buyer's distributional information and the buyer's strategic behavior. We then tailor our framework to the setting of contextual auctions to develop a non-clairvoyant mechanism that achieves no-regret against $\frac{1}{3}$-approximation of the revenue-optimal clairvoyant dynamic mechanism. A natural direction for future work is to improve the regret guarantee or to provide a matching lower bound. Moreover, it is interesting to understand how to apply our framework to dynamic auction environments other than contextual auctions. Finally, it would also be interesting to investigate what can be achieved when the seller has limited prediction power of the future, a region between non-clairvoyant and clairvoyant environments.

## Footnotes

[1]Our results can be extended to multi-buyer settings by using the techniques from Cai et al. [2012] and Mirrokni et al. [2018].

[2]For instance, in a dynamic auction for display advertising, the value of a video ad may be orders of magnitude larger than the value of a text ad.

[3]Interested readers can refer to [Mirrokni et al., 2018] for discussions on the choice of DIC notions.

[4]Note that sublinear revenue loss is only meaningful if the available revenue to extract is itself at least linear, which is the case when $\sum_{t=1}^{T} a_t = \Omega(T)$ since the revenue obtained by the optimal dynamic mechanism is $\Omega(\sum_t a_t)$ in our setting. In fact, a *static* mechanism can already achieve $\Omega(\sum_t a_t)$ revenue by offering a posted price $pa_t$ with $p = 1/(2c_f)$ at stage $t$ which induces revenue at least $p \cdot a_t(1 - p \cdot c_f) = a_t/(4c_f)$ from stage $t$.

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
