[Supplementary Material]

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

**Appendix**

 **A  Helper Lemmas**

365 **Lemma A.1.** *In a single buyer setting, every stage IC and IR mechanism $\langle x, p \rangle$ can be represented*
366 *by a mixture of posted-price auctions such that the probability density to offer a posted price $r$ is*
367 $f(r) = \frac{dx(r)}{dr}$.

368 *Proof.* We show that such a mixture of posted price auctions preserve the allocation rule and payment
369 rule. By the celebrated Myerson's lemma [Myerson, 1981], a mechanism is IC if and only if the
370 allocation rule is monotonically non-decreasing, i.e., $\frac{dx(r)}{dr} \geq 0$ for all valid $r$. Therefore, the density
371 function of posted prices $f(r)$ is well-defined. Moreover, for a buyer with bid $b$, his allocation
372 probability is $\int_0^b f(r)dr = \int_0^b \frac{dx(r)}{dr}dr = x(r)$, which implies that the allocation probability is
373 preserved. Moreover, Myerson's lemma [Myerson, 1981] demonstrated that the payment rule is
374 uniquely determined by the allocation rule: $p(b) = \int_0^b r \cdot \frac{dx(r)}{dr}dr$, which is exactly the payment
375 collected from our mixture of posted price auctions for valuation $v$. □

376 **Lemma A.2.** *For $v \in [\hat{v} - \Delta, \hat{v} + \Delta]$ and any stage IC and IR mechanism $\langle x, p \rangle$, we have*

$$u(v) - \Delta \leq u(\hat{v}) \leq u(v) + \Delta$$

377 *where $u(v) = x(v) \cdot v - p(v)$.*

378 *Proof.* Since $\langle x, p \rangle$ is a stage IC and IR mechanism, by Lemma A.1, we can equivalently offer a
379 mixture of posted price auctions such that the probability density to post a price $r$ is $f(r) = \frac{dx(r)}{dr}$.
380 Therefore, we can express the utility of the buyer for valuation $v_t$ as $\int \frac{dx(r)}{dr}(v - r)^+ dr$. We first
381 show the first inequality:

$$
\begin{aligned}
x(v) \cdot v - p(v) &= \int \frac{dx(r)}{dr}(v - r)^+ dr \\
&\leq \int \frac{dx(r)}{dr}(\hat{v} + \Delta - r)^+ dr \\
&\leq \int \frac{dx(r)}{dr}(\hat{v} - r)^+ dr + \Delta \\
&= x(\hat{v}) \cdot \hat{v} - p(\hat{v}) + \Delta
\end{aligned}
$$

382 where the first equality follows that $\langle x, p \rangle$ is stage-IC and Lemma A.1. By a similar argument, we
383 can prove the second inequality. □

384 The following is a corollary of Lemma A.2, which demonstrates that the difference of expected utility
385 due to the mismatch of distributional information can be related to the estimation error.

386 **Corollary A.1.** *For $F_{(1,T)}$ and $\hat{F}_{(1,T)}$ satisfying Assumption 2, and for any stage IC and IR mecha-*
387 *nism $\langle x, p \rangle$, we have*

$$\mathbb{E}_{v_t \sim F_t}[u(v_t)] - \Delta a_t \leq \mathbb{E}_{v_t \sim \hat{F}_t}[u(v_t)] \leq \mathbb{E}_{v_t \sim F_t}[u(v_t)] + \Delta a_t.$$

388 The following lemma demonstrates that the difference of expected welfare due to the mismatch of
389 distributional information can be related to the estimation error.

390 **Lemma A.3.** *For $F_{(1,T)}$ and $\hat{F}_{(1,T)}$ satisfying Assumption 2, and for any stage IC and IR mechanism*
391 *$\langle x, p \rangle$, we have*

$$\mathbb{E}_{v_t \sim F_t}[x_t(v_t) \cdot v_t] - (c_f + 1)\Delta a_t \leq \mathbb{E}_{v_t \sim \hat{F}_t}[x_t(v_t) \cdot v_t] \leq \mathbb{E}_{v_t \sim F_t}[x_t(v_t) \cdot v_t] + (c_f + 1)\Delta a_t$$

392 *Proof.* Since $\langle x, p \rangle$ is an stage IC and IR mechanism, by Lemma A.1, we can equivalently offer a
393 mixture of posted price auctions such that the probability density to post a price $r$ is $f(r) = \frac{dx(r)}{dr}$.
394 Denote the distribution that $\epsilon_t$ follows as $G_t$.

We first show the first inequality:

$$\mathbb{E}_{v_t \sim F_t}[x_t(v_t) \cdot v_t] = \mathbb{E}_{v_t \sim \hat{F}_t, \epsilon_t \sim G_t}[x_t(v_t + \epsilon_t a_t) \cdot (v_t + \epsilon_t a_t)]$$

$$= \int \frac{dx_t(r)}{dr} \cdot \mathbb{E}_{v_t \sim \hat{F}_t, \epsilon_t \sim G_t}[\mathbf{1}\{v_t + \epsilon_t a_t \geq r\} \cdot (v_t + \epsilon_t a_t)]dr$$

$$\leq \int \frac{dx_t(r)}{dr} \cdot \mathbb{E}_{v_t \sim \hat{F}_t, \epsilon_t \sim G_t}[\mathbf{1}\{v_t + \Delta a_t \geq r\} \cdot (v_t + \Delta a_t)]dr$$

$$\leq \int \frac{dx_t(r)}{dr} \cdot \mathbb{E}_{v_t \sim \hat{F}_t}[\mathbf{1}\{v_t + \Delta a_t \geq r\} \cdot (v_t + \Delta a_t)]dr$$

$$\leq \int \frac{dx_t(r)}{dr} \cdot \mathbb{E}_{v_t \sim \hat{F}_t}[\mathbf{1}\{v_t + \Delta a_t \geq r\} \cdot v_t]dr + \Delta a_t$$

$$= \int \frac{dx_t(r)}{dr} \cdot \mathbb{E}_{v_t \sim \hat{F}_t}[\mathbf{1}\{v_t \geq r\} \cdot v_t + \mathbf{1}\{r - \Delta a_t \leq v_t \leq r\} \cdot v_t]dr + \Delta a_t$$

$$= \mathbb{E}_{v_t \sim \hat{F}_t}[x_t(v_t) \cdot v_t] + \int \frac{dx_t(r)}{dr} \cdot \left[\int_{r-\Delta a_t}^{r} v_t \cdot f_t(v_t)dv_t\right]dr + \Delta a_t$$

$$\leq \mathbb{E}_{v_t \sim \hat{F}_t}[x_t(v_t) \cdot v_t] + (c_f + 1)\Delta a_t \tag{2}$$

where the last inequality follows that $v_t \leq a_t$ and $f_t(v_t) \leq \frac{c_f}{a_t}$. By a similar argument, we can prove the second inequality. $\square$

## B  Omitted Proofs in Section 3

### B.1  Proof of Lemma 3.2

*Proof.* For ease of presentation, let $\psi_t(\mathrm{bal}) = \psi_t(\mathrm{bal}, F_{(1,T)}, F_{(1,T)})$ and $\mathrm{fee}_t(\mathrm{bal}) = \mathrm{fee}_t(\mathrm{bal}, F_t)$. We use a backward induction from $t = T$ to $t = 0$ to show that for all $t$ and $\mathrm{bal} \geq 0$, the inequalities in the statement hold.

For the base case, it is true since for all $\mathrm{bal} \geq 0$, $\psi_T(\mathrm{bal}) = 0$. Assume the induction hypothesis is true for all $t' \geq t$. Then for $t - 1$, first notice that by the computation of $\mathrm{fee}_t(\mathrm{bal})$, we have $\mathrm{fee}_t(\mathrm{bal} + \delta) = \mathrm{fee}_t(\mathrm{bal}) + \delta'$ with $0 \leq \delta' \leq 3\delta$. Therefore, we first have

$$\psi_{t-1}(\mathrm{bal} + \delta) = \mathbb{E}_{v_t \sim F_t}\left[\frac{1}{3}\mathrm{fee}_t(\mathrm{bal} + \delta) + \frac{1}{3}r_t^*(F_t) \cdot \mathbf{1}\{v_t \geq r_t^*(F_t)\} + \psi_t\left(\mathrm{bal} + \delta + \frac{1}{3}v_t - \frac{1}{3}\mathrm{fee}_t(\mathrm{bal} + \delta)\right)\right]$$

$$\geq \mathbb{E}_{v_t \sim F_t}\left[\frac{1}{3}\mathrm{fee}_t(\mathrm{bal}) + \frac{1}{3}r_t^*(F_t) \cdot \mathbf{1}\{v_t \geq r_t^*(F_t)\} + \psi_t\left(\mathrm{bal} + \frac{1}{3}v_t - \frac{1}{3}\mathrm{fee}_t(\mathrm{bal})\right)\right]$$

$$= \psi_{t-1}(\mathrm{bal})$$

where the inequality follows $\delta - \frac{1}{3}\mathrm{fee}_t(\mathrm{bal} + \delta) \geq -\frac{1}{3}\mathrm{fee}_t(\mathrm{bal})$ and the induction hypothesis. We also have

$$\psi_{t-1}(\mathrm{bal} + \delta) = \mathbb{E}_{v_t \sim F_t}\left[\frac{1}{3}\mathrm{fee}_t(\mathrm{bal} + \delta) + \frac{1}{3}r_t^*(F_t) \cdot \mathbf{1}\{v_t \geq r_t^*(F_t)\} + \psi_t\left(\mathrm{bal} + \delta + \frac{1}{3}v_t - \frac{1}{3}\mathrm{fee}_t(\mathrm{bal} + \delta)\right)\right]$$

$$= \mathbb{E}_{v_t \sim F_t}\left[\frac{1}{3}\mathrm{fee}_t(\mathrm{bal}) + \frac{1}{3}r_t^*(F_t) \cdot \mathbf{1}\{v_t \geq r_t^*(F_t)\} + \frac{1}{3}\delta' + \psi_t\left(\mathrm{bal} + \delta + \frac{1}{3}v_t - \frac{1}{3}\mathrm{fee}_t(\mathrm{bal}) - \frac{1}{3}\delta'\right)\right]$$

$$\leq \mathbb{E}_{v_t \sim F_t}\left[\frac{1}{3}\mathrm{fee}_t(\mathrm{bal}) + \frac{1}{3}r_t^*(F_t) \cdot \mathbf{1}\{v_t \geq r_t^*(F_t)\} + \frac{1}{3}\delta' + \psi_t\left(\mathrm{bal} + \frac{1}{3}v_t - \frac{1}{3}\mathrm{fee}_t(\mathrm{bal})\right) + \delta - \frac{1}{3}\delta'\right]$$

$$= \psi_{t-1}(\mathrm{bal}) + \frac{1}{3}\delta' + (\delta - \frac{1}{3}\delta')$$

$$= \psi_{t-1}(\mathrm{bal}) + \delta.$$

where the inequality uses the fact that $\delta - \frac{1}{3}\delta' \geq 0$ and follows the induction hypothesis. $\square$

### B.2  Proof of Lemma 3.3

*Proof.* For ease of presentation, let $\phi_t(\mathrm{bal}) = \psi_t(\mathrm{bal}, \hat{F}_{(1,T)}; \hat{F}_{(1,T)})$ and $\theta_t(\mathrm{bal}) = \psi_t(\mathrm{bal}, F_{(1,T)}; F_{(1,T)})$. We proceed by a backward induction from $t = T$ to $t = 0$ to show that for

all $t$,

$$\phi_t(\mathrm{bal}) \geq \theta_t(\mathrm{bal}) - (\frac{1}{3}c_f + \frac{5}{3})\Delta \sum_{t'=t+1}^{T} a_{t'}.$$

The base case for $t = T$ is clearly true since $\phi_T(\mathrm{bal}) = \theta_T(\mathrm{bal}) = 0$. Assume it is true for all $t' \geq t$ and we consider for stage $t - 1$.

First, if we use $r_t^*(F_t)$ as the reserve price for $\hat{F}_t$, then by Lemma A.3 and Corollary A.1, we have

$$\mathbb{E}_{v_t \sim \hat{F}_t}[r_t^*(\hat{F}_t) \cdot \mathbf{1}\{v_t \geq r_t^*(\hat{F}_t)\}] \geq \mathbb{E}_{v_t \sim \hat{F}_t}[r_t^*(F_t) \cdot \mathbf{1}\{v_t \geq r_t^*(F_t)\}]$$
$$\geq \mathbb{E}_{v_t \sim F_t}[r_t^*(F_t) \cdot \mathbf{1}\{v_t \geq r_t^*(F_t)\}] - (c_f + 2)\Delta a_t. \quad (3)$$

where the first inequality follows that $r_t^*(\hat{F}_t)$ is the Myerson's reserve for $\hat{F}_t$. As for the spend, recall that $\mathrm{fee}_t(\mathrm{bal}; F_t) = \min(3\mathrm{bal}, \mathbb{E}_{v_t \sim F_t}[v_t])$, and thus, we have

$$\mathrm{fee}_t(\mathrm{bal}; F_t) - \Delta a_t \leq \mathrm{fee}_t(\mathrm{bal}; \hat{F}_t) \leq \mathrm{fee}_t(\mathrm{bal}; F_t) + \Delta a_t \quad (4)$$

Therefore, combining (4) and Lemma 3.2, we have

$$\theta_t\left(\mathrm{bal} + \frac{1}{3}(v_t + \epsilon_t a_t) - \frac{1}{3}\mathrm{fee}_t(\mathrm{bal}; \hat{F}_t)\right) \geq \theta_t\left(\mathrm{bal} + \frac{1}{3}v_t - \frac{1}{3}\mathrm{fee}_t(\mathrm{bal}; F_t)\right) - \frac{2}{3}\Delta a_t \quad (5)$$

for $|\epsilon_t| \leq \Delta$. Henceforth, we have

$$\phi_{t-1}(\mathrm{bal}) = \mathbb{E}_{v_t \sim \hat{F}_t}\left[\frac{1}{3}\mathrm{fee}_t(\mathrm{bal}; \hat{F}_t) + \frac{1}{3}r_t^*(\hat{F}_t) \cdot \mathbf{1}\{v_t \geq r_t^*(\hat{F}_t)\} + \phi_t\left(\mathrm{bal} + \frac{1}{3}v_t - \frac{1}{3}\mathrm{fee}_t(\mathrm{bal}; \hat{F}_t)\right)\right]$$

$$\geq \mathbb{E}_{v_t \sim F_t}\left[\frac{1}{3}\mathrm{fee}_t(\mathrm{bal}; F_t) + \frac{1}{3}r_t^*(F_t) \cdot \mathbf{1}\{v_t \geq r_t^*(F_t)\}\right] - (\frac{1}{3}c_f + 1)\Delta a_t$$

$$+ \mathbb{E}_{v_t \sim \hat{F}_t}\left[\theta_t\left(\mathrm{bal} + \frac{1}{3}v_t - \frac{1}{3}\mathrm{fee}_t(\mathrm{bal}; \hat{F}_t)\right)\right] - (\frac{1}{3}c_f + \frac{5}{3})\Delta \sum_{t'=t+1}^{T} a_{t'}$$

$$\geq \mathbb{E}_{v_t \sim F_t}\left[\frac{1}{3}\mathrm{fee}_t(\mathrm{bal}; F_t) + \frac{1}{3}r_t^*(F_t) \cdot \mathbf{1}\{v_t \geq r_t^*(F_t)\}\right]$$

$$+ \mathbb{E}_{v_t \sim F_t}\left[\theta_t\left(\mathrm{bal} + \frac{1}{3}v_t - \frac{1}{3}\mathrm{fee}_t(\mathrm{bal}; F_t)\right)\right] - (\frac{1}{3}c_f + \frac{5}{3})\Delta \sum_{t'=t}^{T} a_{t'}$$

$$= \theta_{t-1}(\mathrm{bal}) - (\frac{1}{3}c_f + \frac{5}{3})\Delta \sum_{t'=t}^{T} a_{t'}$$

where the first inequality follows (3), (4), and the induction hypothesis, and the second inequality follows (5).  $\qquad\square$

## B.3 Proof of Lemma 3.4

*Proof.* We consider a fixed combination of $b_{(1,t)}$ and $v_t$. Let $(X_{t'}, P_{t'})$ be a random variable representing the stage mechanism at stage $t'$. Let $\left(X_{t'}^{OPT}, P_{t'}^{OPT}\right)_{t'=t+1}^{T}$ be the sequence of stage mechanisms corresponding to the optimal play for stages between $t$ and $(T - 1)$ and let $\left(X_{t'}^{Truthful}, P_{t'}^{Truthful}\right)_{t'=t+1}^{T}$ be the sequence of stage mechanisms corresponding to playing truthfully for stages between $t$ and $(T - 1)$.

By playing truthfully for all stages between $t$ and $T$, the buyer's utility is

$$u_t^{Truthful} = \mathbb{E}_{\left(X_{t'}^{Truthful}, P_{t'}^{Truthful}\right)_{t'=t+1}^{T}}\left[\sum_{t'=t+1}^{T} \gamma^{t'-t} \cdot \mathbb{E}_{v_{t'} \sim F_{t'}}\left[v_{t'} \cdot X_{t'}^{Truthful}(v_{t'}) - P_{t'}^{Truthful}(v_{t'})\right]\right].$$

As for the optimal play, the buyer's utility is at most

$$u_t^{OPT} = \mathbb{E}_{\left(X_{t'}^{OPT}, P_{t'}^{OPT}\right)_{t'=t+1}^T} \left[ \sum_{t'=t+1}^T \gamma^{t'-t} \cdot \mathbb{E}_{v_{t'} \sim F_{t'}} \left[ \max_b \left\{ v_{t'} \cdot X_{t'}^{OPT}(b) - P_{t'}^{OPT}(b) \right\} \right] \right]$$

$$= \mathbb{E}_{\left(X_{t'}^{OPT}, P_{t'}^{OPT}\right)_{t'=t+1}^T} \left[ \sum_{t'=t+1}^T \gamma^{t'-t} \cdot \mathbb{E}_{v_{t'} \sim F_{t'}} \left[ v_{t'} \cdot X_{t'}^{OPT}(v_{t'}) - P_{t'}^{OPT}(v_{t'}) \right] \right].$$

where the second equality is due to the fact that the mechanism is stage-IC for $F_{(1,T)}$. Since the mechanism is $\beta_{(1,T)}$-BI, we have

$$U_t(b_{(1,t)}; F_{(1,T)}; \hat{F}_{(1,T)}) \leq u_t^{OPT}$$

$$\leq u_t^{Truthful} + \sum_{t'=t+1}^T \gamma^{t'-t} \cdot \beta_{t'}$$

$$\leq U_t(b_{(1,t-1)}, v_t; F_{(1,T)}; \hat{F}_{(1,T)}) + \sum_{t'=t+1}^T \gamma^{t'-t} \cdot \beta_{t'}.$$

$\square$

## B.4  Proof of Lemma 3.5

*Proof.* By Lemma 3.4, for a buyer who discounts the future with discounting factor $\gamma$, the expected gain in the future by misreporting at round $t$ is at most $\sum_{t'=t+1}^T \gamma^{t'-t} \beta_{t'}$. However, in the random posted-price mechanism at round $t$, the utility loss of a buyer with true valuation $v_t$ from overbidding in a magnitude of $m_t$ is

$$\int_{v_t}^{v_t+m_t} \frac{b - v_t}{a_t} db = \frac{m_t^2}{2a_t}.$$

By a similar calculation, the utility loss of a buyer with true valuation $v_t$ from underbidding in a magnitude of $m_t$ is also $\frac{m_t^2}{2a_t}$. Thus, we have

$$\frac{m_t^2}{2a_t} \leq \sum_{t'=t+1}^T \gamma^{t'-t} \beta_{t'} \Rightarrow m_t \leq \sqrt{\frac{2a_t}{\lambda} \cdot \sum_{t'=t+1}^T \gamma^{t'-t} \beta_{t'}}.$$

$\square$

## B.5  Proof of Lemma 3.6

*Proof.* For ease of presentation, let $\phi_t(\text{bal}) = \psi_t(\text{bal}, \hat{F}_{(1,T)}; F_{(1,T)})$ and $\theta_t(\text{bal}) = \psi_t(\text{bal}, \hat{F}_{(1,T)}; \hat{F}_{(1,T)})$. We proceed by a backward induction from $t = T$ to $t = 0$ to show that for all $t$,

$$\phi_t(\text{bal}) \geq \theta_t(\text{bal}) - (\frac{1}{3}c_f + \frac{5}{3}) \sum_{t'=t+1}^T (\Delta a_{t'} + \eta_{t'})$$

The base case for $t = T$ is clearly true since $\phi_T(\text{bal}) = \theta_T(\text{bal}) = 0$. Assume it is true for all $t' \geq t$ and we consider for stage $t - 1$. First, recall that

$$\phi_{t-1}(\text{bal}) = \mathbb{E}_{v_t \sim F_t} \left[ \frac{1}{3}\text{fee}_t(\text{bal}; \hat{F}_t) + \frac{1}{3}r_t^*(\hat{F}_t) \cdot \mathbf{1}\{v_t' \geq r_t^*(\hat{F}_t)\} + \phi_t \left( \text{bal} + \frac{1}{3}v_t' - \frac{1}{3}\text{fee}_t(\text{bal}; \hat{F}_t) \right) \right]$$

$$= \mathbb{E}_{v_t \sim \hat{F}_t, \epsilon_t} \left[ \frac{1}{3}\text{fee}_t(\text{bal}; \hat{F}_t) + \frac{1}{3}r_t^*(\hat{F}_t) \cdot \mathbf{1}\{v_t' \geq r_t^*(\hat{F}_t)\} + \phi_t \left( \text{bal} + \frac{1}{3}v_t' - \frac{1}{3}\text{fee}_t(\text{bal}; \hat{F}_t) \right) \right]$$

where in the last equality, $v_t$ is the valuation drawn from $\hat{F}_t$ and $v_t'$ is the reported bid given the buyer's true valuation is $v_t + \epsilon_t a_t$ with $|\epsilon_t| \leq \Delta_t$. Therefore, we have $v_t' \in [v_t - \Delta a_t - \eta_t, v_t + \Delta a_t + \eta_t]$. By Lemma A.3 and Corollary A.1,

$$\mathbb{E}_{v_t \sim \hat{F}_t, \epsilon_t}[r_t^*(\hat{F}_t) \cdot \mathbf{1}\{v_t' \geq r_t^*(\hat{F}_t)\}] \geq \mathbb{E}_{v_t \sim \hat{F}_t}[r_t^*(\hat{F}_t) \cdot \mathbf{1}\{v_t \geq r_t^*(\hat{F}_t)\}] - (c_f + 2)(\Delta a_t + \eta_t).$$

(6)

As for the spend, by Lemma 3.2 and the fact that $\text{fee}_t(\text{bal}; \hat{F}_t) \leq \text{fee}_t(\text{bal}; F_t) + \Delta a_t$ we first have

$$\theta_t\left(\text{bal} + \frac{1}{3}v_t' - \frac{1}{3}\text{fee}_t(\text{bal}; \hat{F}_t)\right) \geq \theta_t\left(\text{bal} + \frac{1}{3}v_t - \frac{1}{3}\text{fee}_t(\text{bal}; F_t)\right) - \frac{2}{3}\Delta a_t - \frac{1}{3}\eta_t \quad (7)$$

Henceforth, we have

$$\phi_{t-1}(\text{bal}) = \mathbb{E}_{v_t \sim F_t}\left[\frac{1}{3}\text{fee}_t(\text{bal}; \hat{F}_t) + \frac{1}{3}r_t^*(\hat{F}_t) \cdot \mathbf{1}\{v_t' \geq r_t^*(\hat{F}_t)\} + \phi_t\left(\text{bal} + \frac{1}{3}v_t' - \frac{1}{3}\text{fee}_t(\text{bal}; \hat{F}_t)\right)\right]$$

$$\geq \mathbb{E}_{v_t \sim \hat{F}_t}\left[\frac{1}{3}\text{fee}_t(\text{bal}; \hat{F}_t) + \frac{1}{3}r_t^*(\hat{F}_t) \cdot \mathbf{1}\{v_t \geq r_t^*(\hat{F}_t)\}\right] - (\frac{1}{3}c_f + 1)(\Delta a_t + \eta_t)$$

$$+ \mathbb{E}_{v_t \sim \hat{F}_t}\left[\theta_t\left(\text{bal} + \frac{1}{3}v_t' - \frac{1}{3}\text{fee}_t(\text{bal}; \hat{F}_t)\right)\right] - (\frac{1}{3}c_f + \frac{5}{3})\sum_{t'=t+1}^{T}(\Delta a_{t'} + \eta_{t'})$$

$$\geq \mathbb{E}_{v_t \sim \hat{F}_t}\left[\frac{1}{3}\text{fee}_t(\text{bal}; \hat{F}_t) + \frac{1}{3}r_t^*(\hat{F}_t) \cdot \mathbf{1}\{v_t \geq r_t^*(\hat{F}_t)\}\right]$$

$$+ \mathbb{E}_{v_t \sim \hat{F}_t}\left[\theta_t\left(\text{bal} + \frac{1}{3}v_t - \frac{1}{3}\text{fee}_t(\text{bal}; F_t)\right)\right] - (\frac{1}{3}c_f + \frac{5}{3})\sum_{t'=t}^{T}(\Delta a_{t'} + \eta_{t'})$$

$$= \theta_{t-1}(\text{bal}) - (\frac{1}{3}c_f + \frac{5}{3})\sum_{t'=t}^{T}(\Delta a_{t'} + \eta_{t'})$$

where the first inequality follows (6) and the induction hypothesis, and the second inequality follows (7).

Moreover, notice that we have

$$\sum_t \eta_t = \sum_t \sqrt{\frac{4a_t\Delta}{\lambda} \cdot \sum_{t'=t+1}^{T}\gamma^{t'-t}a_{t'}} = \sqrt{\frac{4\Delta}{\lambda}}\sum_t \sqrt{a_t}\sqrt{\sum_{t'=t+1}^{T}\gamma^{t'-t}a_{t'}}$$

$$\leq \sqrt{\frac{4\Delta}{\lambda}}\sqrt{\sum_t a_t}\sqrt{\sum_t\sum_{t'=t+1}^{T}\gamma^{t'-t}a_{t'}} \leq \sqrt{\frac{4\Delta}{\lambda}}\sqrt{\sum_t a_t}\sqrt{\frac{1}{1-\gamma}\sum_t a_t}$$

$$\leq \sqrt{\frac{4\Delta}{(1-\gamma)\lambda}} \cdot c_a T.$$

where the first inequality follows the Cauchy-Schwarz inequality and the last inequality is due to Assumption 1. $\qquad\square$

## C  Proof of Lemma 4.1

Sketch: Recall that in a contextual auction, the buyer's true valuation is $v_t = a_t(\langle \boldsymbol{\sigma}, \boldsymbol{\zeta}_t\rangle + \varepsilon_t)$ where $a_t$ is the intrinsic value of the item, $\boldsymbol{\zeta}_t$ is the contextual vector, and $\varepsilon_t$ is a random variable following the market noise distribution $M_t$. Notice that $M_t(w_t - \langle \boldsymbol{\sigma}, \boldsymbol{\zeta}_t\rangle)$ is the probability of the event that $\langle \boldsymbol{\sigma}, \boldsymbol{\zeta}_t\rangle + \varepsilon_t = \frac{v_t}{a_t} \leq w_t$, which is equivalent to $v_t \leq a_t \cdot w_t$. As a result, assuming the buyer reports truthfully, $\mathcal{L}_{k-1}(\boldsymbol{\sigma})$ is exactly the negative of log-likelihood corresponding to $\boldsymbol{\sigma}$.

Under truthful reporting, we have $\mathbf{1}\{b_t \geq a_t \cdot w_t\} = \mathbf{1}\{v_t \geq a_t \cdot w_t\}$. For a $\eta_{(1,T)}$-DIC robust dynamic mechanism, we have $v_t - \eta_t \leq b_t \leq v_t + \eta_t$. As a result, if $|a_t \cdot w_t - v_t| > \eta_t$, then any misreport from the buyer does not result in a lie. Therefore, a lie occurs only if the true valuation $v_t \in [a_t \cdot w_t - \eta_t, a_t \cdot w_t + \eta_t]$. By a martingale argument on the sequence of lies, we can obtain that the total number of lies caused by the dynamic mechanism within phase $k$ is $O\left(\sum_{t \in \hat{E}_{k-1}} \frac{\eta_t}{a_t}\right)$. Moreover, the buyer has an additional motivation to misreport to change the seller's estimation for the future phases. However, for $t \in \hat{E}_{k-1}$, the gain from changing the mechanism for the future phases via changing the seller's estimation is relatively small, since the buyer discounts the future.

*Proof.* First, since the mechanism is $\eta_{(1,T)}$-DIC, the misreport within phase $k$ at stage $t$ is bounded by $\eta_t$. We next bound the additional misreport for changing the estimation for the next phase. Note that the utility gain starting from phase $k$ is at most $\sum_{t' \geq \ell_k} \gamma^{t'-t} \cdot a_{t'}$. Under Assumption 1, $a_{t'} \leq c_a \cdot t'$. Therefore, we have for $t \in \hat{E}_{k-1}$,

$$\sum_{t' \geq \ell_k} \gamma^{t'-t} \cdot a_{t'} \leq c_a \cdot \frac{\gamma^{\ell_k - t}}{(1-\gamma)^2} \leq \frac{c_a}{(1-\gamma)^2 \cdot \ell_k^5}$$

Recall that at round $t$, our robust dynamic mechanism is mixed with a random posted price auction with price uniformly drawn from $[0, a_t]$ with probability $\lambda$. Therefore, the additional misreport $\bar{m}_t$ for $t \in \hat{E}_{k-1}$ is at most

$$\lambda_k \cdot \frac{\bar{m}_t^2}{2a_t} \leq \frac{c_a}{(1-\gamma)^2 \cdot \ell_k^5} \Rightarrow \bar{m}_t \leq \sqrt{\frac{2c_a \cdot a_t}{\lambda_k \cdot (1-\gamma)^2 \cdot \ell_k^5}} \leq \sqrt{\frac{2}{\lambda}} \cdot \frac{c_a}{(1-\gamma) \cdot \ell_k^2}$$

where the last inequality is due to $a_t \leq c_a \cdot t \leq c_a \cdot \ell_k$.

To bound the number of lies, for $t \in \hat{E}_{k-1}$, Let $L(j)$ be the number of lies for the first $j$ stages in $\hat{E}_{k-1}$ and $\mathsf{EL}(j)$ be the expected number of lies from stage $(\ell_{k-1} + j)$. Recall that since we sample $w_t$ uniformly from $[0, 1]$ and notice that a lie occurs only if

$$v_t - \eta_t - \bar{m}_t \leq a_t \cdot w_t \leq v_t + \eta_t + \bar{m}_t,$$

which happens with probability at most $2c_f \cdot \frac{\eta_t + \bar{m}_t}{a_t}$. Therefore,

$$\mathsf{EL}(j) \leq 2c_f \cdot \left( \frac{\eta_t}{a_t} + \frac{c'}{\sqrt{\lambda_k} \cdot \ell_k^2} \right).$$

with $t = \ell_{k-1} + j$ and $c' = \frac{\sqrt{2}c_a}{1-\gamma}$. Notice that $\mathbb{E}[L(j) - L(j-1) - \mathsf{EL}(j)] = 0$, which implies that $L(j) - \sum_{j'=0}^{j} \mathsf{EL}(j')$ forms a martingale. Henceforth, by multiplicative Azuma's inequality (see Lemma 10 [Koufogiannakis and Young, 2014]) and denoting $\ell = |\hat{E}_{k-1}|$, we have

$$\Pr[L(\ell) \geq 2(1+\delta) \sum_{j'=0}^{\ell-1} \mathsf{EL}(j')] \leq \exp\left( -\frac{\delta}{2} \cdot \sum_{j'=0}^{\ell-1} \mathsf{EL}(j') \right)$$

By setting $\delta = 2\log \ell_k / \left( \sum_{j'=0}^{\ell-1} \mathsf{EL}(j') \right)$, with probability at least $1 - \frac{1}{\ell_k}$, we have

$$L(\ell) = O\left( \log \ell_k + \sum_{t \in \hat{E}_k} \left( \frac{\eta_t}{a_t} + \frac{1}{\sqrt{\lambda_k} \cdot \ell_k^2} \right) \right) = O\left( \log \ell_k + \sum_{t \in \hat{E}_k} \frac{\eta_t}{a_t} \right).$$

$\square$

# D  Hybrid Non-clairvoyant Mechanism

We adapt $B(\hat{F}_{(1,T)}, \lambda_{(1,K)})$ to obtain a hybrid non-clairvoyant mechanism $B^{hybrid}(\hat{F}_{(1,T)}, \lambda_{(1,K)}, \omega, \tau)$, which is parameterized by a real number $\omega \in (0, 1)$ and a function $\tau : \mathbb{Z}_+ \to \mathbb{R}_+$ that maps the phase number to a real number. The stage mechanism at stage $t$ is parameterized by two non-negative balances $\mathrm{bal}_t$ and $\mathrm{sbal}_t$, and an additional parameter $\mathrm{sw}_t$. In particular, $\mathrm{sw}_t$ is reset to 0 at the beginning of each phase, i.e., for $t = \ell_k$.

For the give-for-free mechanism, the Myerson's auction, and the random posted-price auction, their allocation rules, payment rules, and the update rule for $\mathrm{bal}$ remain the same, while they keep $\mathrm{sw}$ and $\mathrm{sbal}$ the same, i.e., $\mathrm{sw}_{t+1} = \mathrm{sw}_t$ and $\mathrm{sbal}_{t+1} = \mathrm{sbal}_t$. We replace the posted-price auction with extra fee by a hybrid posted-price auction with extra fee.

**Definition D.1** (Hybrid Posted-price Auction with Extra Fee). *For $t \in E_k$, let $E_k^\omega = \{t \mid a_t < \ell_k^\omega\}$.*

- If $t \notin E_k^\omega$: let $\mathsf{fee}_t^b(\mathrm{bal}_t; \hat{F}_t) = \min\left(3\mathrm{bal}_t, \mathbb{E}_{v_t \sim \hat{F}_t}[v_t]\right)$ and $r_t(\mathrm{bal}_t)$ be the posted-price such that
$$\mathbb{E}_{v_t \sim \hat{F}_t}\left[\left(v_t - r_t(\mathrm{bal}_t)\right)^+\right] = \mathsf{fee}_t^b(\mathrm{bal}_t; \hat{F}_t).$$

  The mechanism charges the buyer $\mathsf{fee}_t^b(\mathrm{bal}_t; \hat{F}_t)$ before the buyer learns her valuation and then run a posted-price auction with price $r_t(\mathrm{bal}_t)$
$$x_t^H = \mathbf{1}\{b_t \geq r_t(\mathrm{bal}_t)\},$$
$$p_t^H = \mathsf{fee}_t^b(\mathrm{bal}_t; \hat{F}_t) + r_t(\mathrm{bal}_t) \cdot \mathbf{1}\{b_t \geq r_t(\mathrm{bal}_t)\}$$

  and update the balances: $\mathrm{bal}_{t+1}^H = \mathrm{bal}_t - \mathsf{fee}_t^b(\mathrm{bal}_t; \hat{F}_t)$, $\mathrm{sbal}_{t+1}^H = \mathrm{sbal}_t$, and $\mathrm{sw}_{t+1}^H = \mathrm{sw}_t$.

- otherwise, if $t \in E_k^\omega$: we first update $\mathrm{sw}_{t+1}^H = \mathrm{sw}_t + \mathbb{E}_{v_t \sim \hat{F}_t}[v_t]$;

  - if $\mathrm{sw}_t \geq \tau(k)$, let $\mathsf{fee}_t^s(\mathrm{sbal}_t; \hat{F}_t) = \min\left(3\mathrm{sbal}_t, \mathbb{E}_{v_t \sim \hat{F}_t}[v_t]\right)$ and $r_t(\mathrm{sbal}_t)$ be the posted-price such that
$$\mathbb{E}_{v_t \sim \hat{F}_t}\left[\left(v_t - r_t(\mathrm{sbal}_t)\right)^+\right] = \mathsf{fee}_t^s(\mathrm{sbal}_t; \hat{F}_t)$$

    The mechanism charges the buyer $\mathsf{fee}_t^s(\mathrm{sbal}_t; \hat{F}_t)$ before the buyer learns her valuation and then run a posted-price auction with price $r_t(\mathrm{sbal}_t)$
$$x_t^H = \mathbf{1}\{b_t \geq r_t(\mathrm{sbal}_t)\},$$
$$p_t^H = \mathsf{fee}_t^s(\mathrm{sbal}_t; \hat{F}_t) + r_t(\mathrm{sbal}_t) \cdot \mathbf{1}\{b_t \geq r_t(\mathrm{sbal}_t)\}$$

    and update the balances: $\mathrm{bal}_{t+1}^H = \mathrm{bal}_t$ and
$$\mathrm{sbal}_{t+1}^H = \mathrm{sbal}_t - \mathsf{fee}_t^s(\mathrm{sbal}_t; \hat{F}_t) + \mathbf{1}\{b_t \geq r_t(\mathrm{sbal}_t)\} \cdot (b_t - r_t(\mathrm{sbal}_t));$$

  - otherwise: allocate the item no matter what the buyer's bid is. Moreover, increase the balance $\mathrm{sbal}_t$ by the buyer's bid:
$$x_t^H = 1, \qquad p_t^H = 0,$$
$$\mathrm{bal}_{t+1}^H = \mathrm{bal}_t, \qquad \mathrm{sbal}_{t+1}^H = \mathrm{sbal}_t + b_t.$$

We prove Lemma 4.3 in this section. Lemma 4.3 states that by choosing $\tau(k)$ properly: (1) the revenue loss from running a hybrid non-clairvoyant mechanism against the non-clairvoyant mechanism is small; (2) the number of lies is small. The proof of the first property based on a new revenue tracking program that separates the revenue contribution related to $\mathrm{bal}$ (from stages $t \notin E_k^\omega$) and the revenue contribution related to $\mathrm{sbal}$ (from stages $t \in E_k^\omega$). The argument for the revenue from stages $t \notin E_k^\omega$ simply follows the argument of $\frac{1}{3}$-approximation of the non-clairvoyant mechanism, while the argument for the revenue from stages $t \in E_k^\omega$ exploits the martingale property of $\mathrm{sbal}$ and the fact that $\mathbb{E}_{v_t \sim \hat{F}_t}[v_t]$ is exactly the maximum extra fee we can charge in the posted-price auction with extra fee (Section D.2). We then combine the martingale natural of $\mathrm{sbal}$ and techniques in robust non-clairvoyant mechanism to show the number of lies is small (Section D.3).

## D.1 Bank Account Property

We generalize the definition of BI to accommodate the introduction of $\mathrm{sbal}_t$:

- The mechanism ensures that the expected utility is balance independent if the buyer reports truthfully:
$$\mathbb{E}_{v_t \sim \hat{F}_t}[v_t \cdot x_t(\mathrm{bal}_t, \mathrm{sbal}_t, \mathrm{sw}_t, v_t) - p_t(\mathrm{bal}_t, \mathrm{sbal}_t, \mathrm{sw}_t, v_t)] \tag{sBI}$$

  is a non-negative constant independent of $\mathrm{bal}_t$ and $\mathrm{sbal}_t$.

- A balance update rule never uses more than the total balance from $\text{bal}_t$ and $\text{sbal}_t$, and never deposits more than the buyer's utility into $\text{bal}_t$ and $\text{sbal}_t$ in total:

$$\text{bal}_{t+1} \geq 0, \qquad \text{sbal}_{t+1} \geq 0$$
$$\text{bal}_{t+1} + \text{sbal}_{t+1} \leq \text{bal}_t + \text{sbal}_t + b_t \cdot x_t(\text{bal}_t, \text{sbal}_t, \text{sw}_t, b_t) - p_t^B(\text{bal}_t, \text{sbal}_t, \text{sw}_t, b_t)$$
$$\text{(sBU)}$$

Notice that we allow dependence on $\text{sw}_t$ in sBI. This is because $\text{sw}_t$ is a global parameter such that it is the same at stage $t$ for all possible historical bids in the past.

**Lemma D.1.** *The hybrid non-clairvoyant mechanism* $B^{hybrid}(\hat{F}_{(1,T)}, \lambda_{(1,K)}, \omega, \tau)$ *is stage-IC, sBI and sBU for* $\hat{F}_{(1,T)}$. *Therefore,* $B^{hybrid}(\hat{F}_{(1,T)}, \lambda_{(1,K)}, \omega, \tau)$ *is* $\eta_{(1,T)}$*-DIC with* $\eta_t = 0$ *and ex-post IR for* $\hat{F}_{(1,T)}$.

*Proof.* Since all mechanisms are variants of leave-it-or-take-it mechanisms, the mixture of them is clearly stage-IC. For sBU, notice that we only decrease $\text{bal}$ and $\text{sbal}$ in the posted price auction, and moreover, by the construction of $r_t(\text{bal}_t, \text{sbal}_t; \hat{F}_t)$, it is at most $\text{bal}_t$ ($\text{sbal}_t$) when $\text{bal}_t$ ($\text{sbal}_t$) is deducted. Furthermore, it is straightforward to verify that the sum of the deposit to $\text{bal}$ and $\text{sbal}$ is at most the buyer's utility at stage $t$. Therefore, the mechanism is sBU. To demonstrate the mechanism is sBI, notice that when $a_t \geq \ell_k^\omega$ or $\text{sw}_t \geq \tau(k)$, the buyer's expected utility is exactly $\mathbb{E}_{v_t \sim \hat{F}_t}[v_t] + 0 + \mathbb{E}_{v_t \sim \hat{F}_t}[(v_t - r_t^*(\hat{F}_t)]$ for all $\text{bal}_t$ and $\text{sbal}_t$; otherwise, the buyer's expected utility is $2\mathbb{E}_{v_t \sim \hat{F}_t}[v_t] + \mathbb{E}_{v_t \sim \hat{F}_t}[(v_t - r_t^*(\hat{F}_t)]$ for all $\text{bal}_t$ and $\text{sbal}_t$. Thus, the mechanism is sBI.

Since the mechanism is sBI, the buyer's historical reports have no impact on her future expected utilities, assuming she reports truthfully in the future. Combining with the fact that the mechanism is stage-IC for every stage, the mechanism is $\eta_{(1,T)}$-DIC with $\eta_{(1,T)} = (0, \ldots, 0)$. Moreover, by the balance update property sBU, the nonnegative $\text{bal}_t + \text{sbal}_t$ always lower bounds the buyer's utility provided truthful reporting. Thus, the mechanism is ex-post IR. $\qquad\square$

## D.2 Revenue Tracking Program

We develop a program to compute the revenue obtained from the hybrid non-clairvoyant mechanism. For convenience, let $\text{fee}_t^b(\text{bal}; \hat{F}_t) = 0$ for $t \in E_k^\omega$ (in which $\text{fee}_t^b(\text{bal}; \hat{F}_t)$ is not defined). Moreover, for stage $t$ such that $t \in E_k^\omega$ and $\text{sw}_t < \tau(k)$ (in which $\text{fee}_t^s(\text{sbal}; \hat{F}_t)$ and $r_t(\text{sbal})$ are not defined), let $\text{fee}_t^s(\text{sbal}; \hat{F}_t) = r_t(\text{sbal}) = 0$.

**Definition D.2.** *For a hybrid non-clairvoyant mechanism* $B^{hybrid}(\hat{F}_{(1,T)}, \lambda_{(1,K)}, \omega, \tau)$, *we consider revenue tracking programs* $\psi_t^b(\text{bal}, \hat{F}_{(1,T)}; F_{(1,T)})$ *and* $\psi_t^s(\text{sbal}, \hat{F}_{(1,T)}; F_{(1,T)})$ *to keep track on the revenue of implementing* $B^{hybrid}(\hat{F}_{(1,T)}, \lambda_{(1,K)}, \omega, \tau)$ *when the buyer's true distribution is* $F_{(1,T)}$. *We define* $\psi_T^b(\text{bal}) = \psi_T^s(\text{sbal}) = 0$ *and for* $t < T$,

$$\psi_{t-1}^b(\text{bal}, \hat{F}_{(1,T)}; F_{(1,T)}) = \mathbb{E}_{v_t \sim F_t} \left[ \frac{1}{3}\text{fee}_t^b(\text{bal}; \hat{F}_t) + \frac{1}{3}r_t^*(\hat{F}_t) \cdot \mathbf{1}\{v_t' \geq r_t^*(\hat{F}_t)\} \right.$$
$$\left. + \psi_t^b \left( \text{bal} + \frac{1}{3}v_t' - \frac{1}{3}\text{fee}_t^b(\text{bal}; \hat{F}_t), \hat{F}_{(1,T)}; F_{(1,T)} \right) \right] \quad (8)$$

*where* $v_t'$ *is the buyer's reported bid that maximizes her continuation utility when the buyer's true valuation is* $v_t$.

*Moreover, for* $t \notin E_k^\omega$, $\psi_{t-1}^s(\text{sbal}, \hat{F}_{(1,T)}; F_{(1,T)}) = \psi_t^s(\text{sbal}, \hat{F}_{(1,T)}; F_{(1,T)})$; *otherwise,*

$$\psi_{t-1}^s(\text{sbal}, \hat{F}_{(1,T)}; F_{(1,T)})$$
$$= \mathbb{E}_{v_t \sim F_t} \left[ \frac{1}{3}\text{fee}_t^s(\text{sbal}; \hat{F}_t) + \psi_t^s \left( \text{sbal} + \frac{1}{3}\left(v_t' - \text{fee}_t^s(\text{sbal}; \hat{F}_t) - r_t(\text{sbal})\right), \hat{F}_{(1,T)}; F_{(1,T)} \right) \right]$$
$$(9)$$

*where* $v_t'$ *is the reported bid when the buyer's true valuation is* $v_t$.

561 Notice that we separate the revenue tracking for $\mathrm{bal}$ and $\mathrm{sbal}$. Moreover, the revenue obtained
562 from the Myerson's auction are counted in $\psi_t^b$. Similar to the revenue tracking program for the
563 non-clairvoyant mechanism (1), we record the revenue from each stage $t$ while omitting the possible
564 revenue $r_t(\mathrm{bal})$ or $r_t(\mathrm{sbal})$ from the posted-price auction with extra fee.

565 **Proposition D.1.** $\mathrm{Rev}\left(B^{hybrid}(\hat{F}_{(1,T)}, \lambda_{(1,K)}, \omega, \tau); F_{(1,T)}\right) \geq \psi_0^b(0, \hat{F}_{(1,T)}; F_{(1,T)}) +$
566 $\psi_0^s(0, \hat{F}_{(1,T)}; F_{(1,T)}) - O(\lambda T)$.

567 **Revenue Performance with Perfect Distributional Information**   We first compare the revenue
568 obtained by the hybrid non-clairvoyant mechanism and the non-clairvoyant mechanism, when the
569 seller's distributional information is perfect, i.e., $\hat{F}_{(1,T)} = F_{(1,T)}$. Notice that the definition of $\psi_t^b$ (8)
570 is exactly the same as $\psi_t$ for the non-clairvoyant mechanism (1). However, the difference is that for
571 stage $t$ in which $a_t < \ell_k^\omega$, $\mathrm{fee}_t^b(\mathrm{bal}; \hat{F}_t) = 0$ in the hybrid non-clairvoyant mechanism. Following an
572 argument similar to the proof of Lemma 3.2, we have the following lemma:

573 **Lemma D.2.** *For any* $F_{(1,T)}$*, we have for all* $0 \leq t \leq T$*,*

$$\psi_t^b(\mathrm{bal} + \delta, F_{(1,T)}; F_{(1,T)}) - \delta \leq \psi_t^b(\mathrm{bal}, F_{(1,T)}; F_{(1,T)}) \leq \psi_t^b(\mathrm{bal} + \delta, F_{(1,T)}; F_{(1,T)}).$$

574 Therefore, all our results for the robust non-clairvoyant mechanism (Section 3) works for the rev-
575 enue obtained from $\psi_t^b$. We then compute the revenue obtain from $\psi_t^b$ when the seller has perfect
576 distributional information.

577 **Lemma D.3.** $\psi_0^b(0, F_{(1,T)}; F_{(1,T)}) \geq \psi_0(0, F_{(1,T)}; F_{(1,T)}) - \frac{1}{3}\sum_k \sum_{t \in E_k^\omega} \mathbb{E}_{v_t \sim F_t}[v_t]$.

578 *Proof.* For simplicity, let $\phi_t(\mathrm{bal}) = \psi_t^b(\mathrm{bal}, F_{(1,T)}; F_{(1,T)})$ and $\theta_t(\mathrm{bal}) = \psi_t(\mathrm{bal}, F_{(1,T)}; F_{(1,T)})$.
579 We prove by a backward induction from $t = T$ to $t = 0$ to show that for all $t$ and $\mathrm{bal} \geq 0$,

$$\phi_t(\mathrm{bal}) \geq \theta_t(\mathrm{bal}) - \frac{1}{3}\sum_k \sum_{t' \in E_k^\omega, t' > t} \mathbb{E}_{v_{t'} \sim F_{t'}}[v_{t'}].$$

580 The base case is true for $t = T$ since $\phi_T(\mathrm{bal}) = \theta_T(\mathrm{bal}) = 0$ for all $\mathrm{bal} \geq 0$. Assume the
581 induction hypothesis is true for $t' \geq t$ and we consider stage $t - 1$. For $t \in E_k$, if $t \notin E_k^\omega$, we have
582 $\mathrm{fee}_t^b(\mathrm{bal}; F_t) = \mathrm{fee}_t(\mathrm{bal}; F_t)$. Therefore, we have

$$\phi_{t-1}(\mathrm{bal}) = \mathbb{E}_{v_t \sim F_t}\left[\frac{1}{3}\mathrm{fee}_t^b(\mathrm{bal}; F_t) + \frac{1}{3}r_t^*(F_t) \cdot \mathbf{1}\{v_t \geq r_t^*(F_t)\} + \phi_t\left(\mathrm{bal} + \frac{1}{3}v_t - \frac{1}{3}\mathrm{fee}_t^b(\mathrm{bal}; F_t)\right)\right]$$

$$= \mathbb{E}_{v_t \sim F_t}\left[\frac{1}{3}\mathrm{fee}_t(\mathrm{bal}; F_t) + \frac{1}{3}r_t^*(F_t) \cdot \mathbf{1}\{v_t \geq r_t^*(F_t)\} + \phi_t\left(\mathrm{bal} + \frac{1}{3}v_t - \frac{1}{3}\mathrm{fee}_t(\mathrm{bal}; F_t)\right)\right]$$

$$= \theta_{t-1}(\mathrm{bal}) + \mathbb{E}_{v_t \sim F_t}\left[\phi_t\left(\mathrm{bal} + \frac{1}{3}v_t - \frac{1}{3}\mathrm{fee}_t(\mathrm{bal}; F_t)\right) - \theta_t\left(\mathrm{bal} + \frac{1}{3}v_t - \frac{1}{3}\mathrm{fee}_t(\mathrm{bal}; F_t)\right)\right]$$

$$\geq \theta_{t-1}(\mathrm{bal}) - \frac{1}{3}\sum_k \sum_{t' \in E_k^\omega, t' \geq t} \mathbb{E}_{v_{t'} \sim F_{t'}}[v_{t'}]$$

583 where the inequality follows the induction hypothesis. On the other hand, if $t \in E_k^\omega$, we have
584 $\mathrm{fee}_t^b(\mathrm{bal}; F_t) = 0$. As a result, we have

$$\phi_{t-1}(\mathrm{bal}) = \mathbb{E}_{v_t \sim F_t}\left[\frac{1}{3}\mathrm{fee}_t^b(\mathrm{bal}; F_t) + \frac{1}{3}r_t^*(F_t) \cdot \mathbf{1}\{v_t \geq r_t^*(F_t)\} + \phi_t\left(\mathrm{bal} + \frac{1}{3}v_t - \frac{1}{3}\mathrm{fee}_t^b(\mathrm{bal}; F_t)\right)\right]$$

$$\geq \mathbb{E}_{v_t \sim F_t}\left[\frac{1}{3}r_t^*(F_t) \cdot \mathbf{1}\{v_t \geq r_t^*(F_t)\} + \phi_t\left(\mathrm{bal} + \frac{1}{3}v_t - \mathrm{fee}_t(\mathrm{bal}; F_t)\right)\right]$$

$$= \theta_{t-1}(\mathrm{bal}) - \frac{1}{3}\mathrm{fee}_t(\mathrm{bal}; F_t)$$

$$\quad + \mathbb{E}_{v_t \sim F_t}\left[\phi_t\left(\mathrm{bal} + \frac{1}{3}v_t - \frac{1}{3}\mathrm{fee}_t(\mathrm{bal}; F_t)\right) - \theta_t\left(\mathrm{bal} + \frac{1}{3}v_t - \frac{1}{3}\mathrm{fee}_t(\mathrm{bal}; F_t)\right)\right]$$

$$\geq \theta_{t-1}(\mathrm{bal}) - \frac{1}{3}\sum_k \sum_{t' \in E_k^\omega, t' \geq t} \mathbb{E}_{v_{t'} \sim F_{t'}}[v_{t'}]$$

where the first inequality follows Lemma D.2 and the second inequality uses the induction hypothesis and the fact that $\mathsf{fee}_t(\mathrm{bal}; F_t) \le \mathbb{E}_{v_t \sim F_t}[v_t]$. $\qquad\square$

Let $\tilde{E}_k^\omega = \{t \in E_k^\omega \mid \mathrm{sw}_t < \tau(k)\}$. Let $t^*(k) = \max \tilde{E}_k^\omega$ and consider a sequence $y_t$ for $t \in E_k^\omega$ such that

$$
y_t = \begin{cases} \frac{1}{3} \sum_{t' \in E_k, t' \le t} E_{v_{t'} \sim F_{t'}}[v_{t'}] & t' \in \tilde{E}_k^\omega \\ y_{t^*(k)} & t' \notin \tilde{E}_k^\omega \end{cases}
$$

Henceforth, the key observation is that the sequence $\{\mathrm{sbal}_t - y_t\}_{t \in E_k^\omega}$ forms a martingale with bounded difference $a_t$ at stage $t$: for $t \in \tilde{E}_k^\omega$, we have

$$
\mathbb{E}_{v_t \sim F_t}[\mathrm{sbal}_{t+1} - \mathrm{sbal}_t] = \mathbb{E}_{v_t \sim F_t}[\mathrm{sbal}_t + \tfrac{1}{3} v_t - \mathrm{sbal}_t] = \frac{1}{3} \mathbb{E}_{v_t \sim F_t}[v_t]
$$

and for $t \in E_k^\omega \setminus \tilde{E}_k^\omega$, we have

$$
\mathbb{E}_{v_t \sim F_t}[\mathrm{sbal}_{t+1} - \mathrm{sbal}_t] = \mathbb{E}_{v_t \sim F_t}\left[ \mathrm{sbal}_t + \frac{1}{3}\left( v_t - \mathsf{fee}_t^s(\mathrm{sbal}_t; F_t) - r_t(\mathrm{sbal}_t) \right) - \mathrm{sbal}_t \right] = 0
$$

where the last equality follows the fact that $\mathbb{E}_{v_t \sim \hat{F}_t}\left[ (v_t - r_t(\mathrm{sbal}_t))^+ \right] = \mathsf{fee}_t^s(\mathrm{sbal}_t; \hat{F}_t)$ from the construction of the hybrid non-clairvoyant mechanism.

**Lemma D.4.** *If $\tau(k) \ge 4\sqrt{c_a} \cdot \ell_k^{\frac{1}{2}(1+\omega)} \sqrt{\log \ell_k}$, for any $t \in E_k^\omega \setminus \tilde{E}_k^\omega$, $\Pr\left[ \mathrm{sbal}_t < y_{t^*(k)} - \delta \right] \le \exp\left( -\frac{\delta^2}{4c_a \ell_k^{1+\omega}} \right).$*

*Proof.* Notice that $y_{t^*(k)} \ge \tau(k)$ and by Azuma's inequality, we have for any $t \in E_k^\omega \setminus \tilde{E}_k^\omega$,

$$
\Pr\left[ \mathrm{sbal}_t < y_{t^*(k)} - \delta \right] \le \exp\left( -\frac{\delta^2}{2\sum_{t \in E_k^\omega} a_t^2} \right) \le \exp\left( -\frac{\delta^2}{4c_a \ell_k^{1+\omega}} \right)
$$

where the second inequality follows that

$$
\sum_{t \in E_k^\omega} a_t^2 \le (\ell_k^\omega)^2 \cdot \frac{\sum_{t \in E_k^\omega} a_t}{\ell_k^\omega} \le (\ell_k^\omega)^2 \cdot \frac{2c_a \ell_k}{\ell_k^\omega} = 2c_a \cdot \ell_k^{1+\omega}
$$

where the second inequality follows Assumption 1. $\qquad\square$

**Lemma D.5.** *If $\tau(k) \ge 4\sqrt{c_a} \cdot \ell_k^{\frac{1}{2}(1+\omega)} \sqrt{\log \ell_k}$, we have*

$$
\psi_0^s(0, F_{(1,T)}; F_{(1,T)}) \ge \frac{1}{3} \sum_k \left( \sum_{t \in E_k^\omega} \mathbb{E}_{v_t \sim F_t}[v_t] - \tau(k) \right) - \tilde{O}(T^\omega).
$$

*Proof.* For convenience, let $\phi_t(\mathrm{sbal}) = \psi_t^s(\mathrm{sbal}, F_{(1,T)}; F_{(1,T)})$. Recall that $\phi_{t-1}(\mathrm{sbal}) = \phi_t(\mathrm{sbal})$ if $t \in E_k$ and $t \notin E_k^\omega$. Moreover, recall that $\mathrm{sw}_t$ is set to 0 at stage $\ell_k$ for all $k$ and when $t \in E_k$ and $\mathrm{sw}_t < \tau(k)$, we in fact offer a give-for-free mechanism in the hybrid posted-price auction. Therefore, the mechanism does not accrue any revenue from stages with $t \in \tilde{E}_k^\omega$.

Plugging in $\delta = \tau(k) - \frac{1}{3}\ell_k^\omega$ in Lemma D.4, we have

$$
\Pr\left[ \mathrm{sbal}_t < \frac{1}{3}\ell_k^\omega \right] \le \Pr\left[ \mathrm{sbal}_t < y_{t^*(k)} - \tau(k) + \frac{1}{3}\ell_k^\omega \right] \le \exp\left( -\frac{\left( \tau(k) - \frac{1}{3}\ell_k^\omega \right)^2}{4c_a \ell_k^{1+\omega}} \right) \le \frac{1}{\ell_k^2}
$$

where the first inequality is due to $y_{t^*(k)} \ge \tau(k)$. Applying the union bound, we have

$$
\Pr\left[ \exists t \in E_k^\omega \setminus \tilde{E}_k^\omega, \mathrm{sbal}_t < \frac{1}{3}\ell_k^\omega \right] \le \frac{1}{\ell_k}.
$$

Therefore, with probability at least $(1 - \frac{1}{\ell_k})$, for all $t \in E_k^\omega \setminus \tilde{E}_k^\omega$, $3\mathrm{sbal}_t \geq \ell_k^\omega \geq a_t \geq \mathbb{E}_{v_t \sim F_t}[v_t]$, which implies that $\mathrm{fee}_t^s(\mathrm{sbal}_t; F_t) = \mathbb{E}_{v_t \sim F_t}[v_t]$. Thus, combining with the fact that $y_{t^*(k)} \leq \tau(k) + \ell_k^\omega$ for all $k$, we have for the revenue obtained from $\frac{1}{3}\mathrm{fee}_t^s(\mathrm{sbal}_t; F_t)$ is at least

$$(1 - \frac{1}{\ell_k}) \cdot \frac{1}{3}\left(\sum_{t \in E_k^\omega} \mathbb{E}_{v_t \sim F_t}[v_t] - y_{t^*(k)}\right) = \frac{1}{3}\left(\sum_{t \in E_k^\omega} \mathbb{E}_{v_t \sim F_t}[v_t] - \tau(k)\right) - O(\ell_k^\omega)$$

We conclude the proof of the lemma by taking the summation over all the phases. $\qquad\square$

Combining Lemma D.3 and D.5, we can conclude that

**Corollary D.1.** *By setting $\tau(k) = 4\sqrt{c_a} \cdot \ell_k^{\frac{1}{2}(1+\omega)}\sqrt{\log \ell_k}$, we have*

$$\psi_0^b(0, F_{(1,T)}; F_{(1,T)}) + \psi_0^s(0, F_{(1,T)}; F_{(1,T)}) \geq \psi_0(0, F_{(1,T)}; F_{(1,T)}) - \tilde{O}\left(T^{\frac{1}{2}(1+\omega)}\right).$$

Therefore, the revenue loss of the hybrid non-clairvoyant mechanism against the optimal clairvoyant mechanism is sublinear in $T$ when $\omega \in (0, 1)$.

### D.3  Analysis on the Misreport

We analyze the buyer's misreport in this section. By the discussion in Section 4.1, we focus on $\hat{E}_k$ instead of $E_k$. We first provide a naive bound for the property of $\eta_{(1,T)}$-DIC in $\hat{E}_k$ for $B^{hybrid}(\hat{F}_{(1,T)}, \lambda_{(1,K)}, \omega, \tau)$.

**Proposition D.2.** *In $B^{hybrid}(\hat{F}_{(1,T)}, \lambda_{(1,K)}, \omega, \tau)$, for $t \in \hat{E}_k$, we have*

$$\eta_t \leq 4\sqrt{\frac{a_t \Delta_k}{\lambda_k} \cdot \sum_{t' \in E_k, t' > t} \gamma^{t'-t}a_{t'}}$$

*and moreover,*

$$\sum_{t \in \hat{E}_k} \eta_t = 4c_a\sqrt{\frac{\Delta_k}{(1-\gamma)\lambda_k}} \cdot \ell_k.$$

*Proof.* By Lemma 3.5, we have

$$
\begin{aligned}
\eta_t &\leq \sqrt{\frac{4a_t \Delta_k}{\lambda_k} \cdot \sum_{t'=t+1}^{T} \gamma^{t'-t}a_{t'}} \\
&= \eta_t \leq \sqrt{\frac{4a_t \Delta_k}{\lambda_k} \cdot \left(\sum_{t' \in E_k, t' > t} \gamma^{t'-t}a_{t'} + \sum_{t' \geq \ell_{k+1}} \gamma^{t'-t}a_{t'}\right)} \\
&\leq \eta_t \leq \sqrt{\frac{4a_t \Delta_k}{\lambda_k} \cdot \left(\sum_{t' \in E_k, t' > t} \gamma^{t'-t}a_{t'} + \frac{c_a}{(1-\gamma)^2 \cdot \ell_{k+1}^5}\right)} \\
&\leq \eta_t \leq 4\sqrt{\frac{a_t \Delta_k}{\lambda_k} \cdot \left(\sum_{t' \in E_k, t' > t} \gamma^{t'-t}a_{t'}\right)}
\end{aligned}
$$

Combining with the argument similar to the proof of Lemma 3.6, we can conclude that

$$\sum_{t \in \hat{E}_k} \eta_t = 4c_a\sqrt{\frac{\Delta_k}{(1-\gamma)\lambda_k}} \cdot \ell_k.$$

$\qquad\square$

For convenience, let $\hat{E}_k^\omega = E_k^\omega \cap \hat{E}_k$ and let

$$A_k^\omega = \{t \in \hat{E}_k^\omega \mid \mathsf{next}(t) > 6\log_{1/\gamma} \ell_k\}$$

where $\mathsf{next}(t) = \min\left(\{t' > t \mid t' \in E_k \setminus E_k^\omega\} \cup \{\ell_{k+1}\}\right) - t'$. Intuitively, $\mathsf{next}(t)$ is the distance between stage $t$ and the first future stage not in $E_k^\omega$. Henceforth, $A_k^\omega$ is a set of stages in which the first future stage not in $E_k^\omega$ is at least $6\log_{1/\gamma} \ell_k$ far away. By Lemma D.4, with probability at least $(1 - \frac{1}{\ell_k})$, the mechanism we offer in $E_k^\omega$ is static, which implies that the buyer has little incentive to misreport for stages in $A_k^\omega$ since she discounts the future. We formalize this intuition in Lemma D.6. For convenience, let

$$\tau^{\Delta_k, \lambda_k}(k) = 4\sqrt{c_a} \cdot \ell_k^{\frac{1}{2}(1+\omega)} \sqrt{\log \ell_k} + 5c_a \sqrt{\frac{\Delta_k}{(1-\gamma)\lambda_k}} \cdot \ell_k + 6\ell_k^\omega \log_{1/\gamma} \ell_k.$$

**Lemma D.6.** *In $B^{hybrid}(\hat{F}_{(1,T)}, \lambda_{(1,K)}, \omega, \tau)$, if $\tau(k) \geq \tau^{\Delta_k, \lambda_k}(k)$ then with probability at least $(1 - \frac{1}{\ell_k})$, for all $t \in A_k^\omega$, $\eta_t \leq O\left(\frac{1}{\sqrt{\lambda_k} \cdot \ell_k^2}\right)$ and moreover,*

$$\psi_{\ell_k}^s(0, \hat{F}_{(\ell_k, \ell_{k+1}-1)}; F_{(\ell_k, \ell_{k+1}-1)}) \geq \frac{1}{3}\left(\sum_{t \in E_k^\omega} \mathbb{E}_{v_t \sim \hat{F}_t}[v_t] - \tau(k)\right) - O(\ell_k^\omega).$$

*Proof.* First, plugging $\delta = \tau(k) - 5c_a\sqrt{\frac{\Delta_k}{(1-\gamma)\lambda_k}} \cdot \ell_k - 6\ell_k^\omega \log_{1/\gamma} \ell_k - \frac{1}{3}\ell_k^\omega$ into Lemma D.4, then conditioned on the assumption that the buyer reports truthfully according to $\hat{F}_{(1,T)}$, we have

$$\Pr\left[\mathsf{sbal}_t < 5c_a\sqrt{\frac{\Delta_k}{(1-\gamma)\lambda_k}} \cdot \ell_k + 6\ell_k^\omega \log_{1/\gamma} \ell_k + \frac{1}{3}\ell_k^\omega\right]$$

$$\leq \Pr\left[\mathsf{sbal}_t < y_{t^*(k)} - \left(\tau(k) - 5c_a\sqrt{\frac{\Delta_k}{(1-\gamma)\lambda_k}} \cdot \ell_k - 6\ell_k^\omega \log_{1/\gamma} \ell_k - \frac{1}{3}\ell_k^\omega\right)\right]$$

$$\leq \exp\left(-\frac{\left(5\sqrt{c_a} \cdot \ell_k^{\frac{1}{2}(1+\omega)} \sqrt{\log \ell_k} - \frac{1}{3}\ell_k^\omega\right)^2}{4c_a \ell_k^{1+\omega}}\right) \leq \frac{1}{\ell_k^2}.$$

Applying a union bound, we have

$$\Pr\left[\exists t \in A_k^\omega, \, \mathsf{sbal}_t < 5c_a\sqrt{\frac{\Delta_k}{(1-\gamma)\lambda_k}} \cdot \ell_k + 6\ell_k^\omega \log_{1/\gamma} \ell_k + \frac{1}{3}\ell_k^\omega\right] \leq \frac{1}{\ell_k}$$

conditioned on the assumption that the buyer reports truthfully according to $\hat{F}_{(1,T)}$. If the buyer misreports under $F_{(1,T)}$, then by Proposition D.2 and Assumption 1, we have

$$\Pr\left[\exists t \in A_k^\omega, \, \mathsf{sbal}_t < 6\ell_k^\omega \log_{1/\gamma} \ell_k + \frac{1}{3}\ell_k^\omega\right] \leq \frac{1}{\ell_k}.$$

By the definition of $A_k^\omega$, for $t' \in \left[t+1, t+6\log_{1/\gamma} \ell_k\right]$, the mechanism at stage $t'$ is the same if $\mathsf{sbal}_{t'} \geq \frac{1}{3}\ell_k^\omega$, which implies that $3\mathsf{sbal}_{t'} \geq a_{t'} \geq \mathbb{E}_{v_t \sim \hat{F}_t}[v_t]$. Moreover, notice that at stage $t'$, since $a_{t'} \leq \ell_k^\omega$, the decrement of $\mathsf{sbal}$ is at most $\ell_k^\omega$. Therefore, we have

$$\Pr\left[\exists t \in A_k^\omega, \exists t < t' \leq t+6\log_{1/\gamma} \ell_k, \, \mathsf{sbal}_t < \frac{1}{3}\ell_k^\omega\right] \leq \frac{1}{\ell_k}.$$

As a result, for a buyer who misreports at stage $t \in A_k^\omega$, she can only earn benefit from stages at least $6\log_{1/\gamma} \ell_k$ away in the future. Thus, the amount of misreport $m_t$ must satisfy

$$\lambda_k \cdot \frac{m_t^2}{2a_t} \leq \sum_{t' \geq t+6\log_{1/\gamma} \ell_k} \gamma^{t'-t} a_{t'} \leq \frac{c_a}{(1-\gamma)^2 \cdot \ell_k^5} \Rightarrow m_t = O\left(\frac{1}{\sqrt{\lambda_k} \cdot \ell_k^2}\right).$$

642 For the revenue guarantee in phase $k$, notice that our analysis demonstrates that once $\mathrm{sw}_t \geq \tau(k) \geq$
643 $\tau^{\Delta_k, \lambda_k}(k)$ for some $t$, even the buyer misreports, with probability at least $(1 - \frac{1}{\ell_k})$, $\mathrm{sbal}_{t'} \geq \frac{1}{3}\ell_k^\omega$
644 for all $t' \geq t$ and $t' \in E_k^\omega$. Therefore, the mechanism can obtain revenue $\frac{1}{3}\mathbb{E}_{v_{t'} \sim \hat{F}_{t'}}[v_{t'}]$ for these
645 stages. $\qquad\square$

646 We are now ready to bound the estimation error of our learning policy (Section 4.1) in our robust
647 hybrid non-clairvoyant bank account mechanism $B^{hybrid}(\hat{F}_{(1,T)}, \lambda_{(1,K)}, \omega, \tau)$.

648 **Lemma D.7.** *If* $\tau(k) \geq \tau^{\Delta_k, \lambda_k}(k)$ *and* $\lambda_k \geq \ell_k^{-2}$, *then with probability at least* $(1 - \frac{1}{\ell_k})$,
649 $B^{hybrid}(\hat{F}_{(1,T)}, \lambda_{(1,K)}, \omega, \tau)$ *is* $\eta_{(1,T)}$*-DIC for* $F_{(1,T)}$ *such that*

$$\sum_{t \in \hat{E}_k} \frac{\eta_t}{a_t} \leq \tilde{O}\left(\ell_k^{1-\omega}\right).$$

650 *Proof.* First of all, for $t \in A_k^\omega$, by Lemma D.6, we have

$$\sum_{t \in A_k^\omega} \frac{\eta_t}{a_t} \leq \sum_{t \in A_k^\omega} \eta_t = O\left(\frac{1}{\sqrt{\lambda_k} \cdot \ell_k}\right)$$

651 where the inequality is due to $a_t \geq 1$. As for $t \in \hat{E}_k^\omega \setminus A_k^\omega$, we simply apply the bound $\frac{\eta_t}{a_t} \leq 1$ since
652 $\eta_t \leq a_t$. Moreover, by the definition of $A_k^\omega$, for $t \in \hat{E}_k^\omega$, there are at most $|E_k \setminus E_k^\omega| \cdot 6 \log_{1/\gamma} \ell_k$
653 stages not in $A_k^\omega$. In addition, by Assumption 1, we have $|E_k \setminus E_k^\omega| = O(\ell_k^{1-\omega})$. Therefore, we have

$$\sum_{t \in \hat{E}_k^\omega \setminus A_k^\omega} \frac{\eta_t}{a_t} \leq |\hat{E}_k^\omega \setminus A_k^\omega| = \tilde{O}\left(\ell_k^{1-\omega}\right).$$

654 Finally, for stages in $\hat{E}_k \setminus \hat{E}_k^\omega$, we have $|\hat{E}_k \setminus \hat{E}_k^\omega| = O(\ell_k^{1-\omega})$, and therefore,

$$\sum_{t \in \hat{E}_k \setminus \hat{E}_k^\omega} \frac{\eta_t}{a_t} \leq |\hat{E}_k \setminus \hat{E}_k^\omega| = \tilde{O}\left(\ell_k^{1-\omega}\right).$$

655 We conclude the proof of the lemma by summing over all three cases. $\qquad\square$