[Reviews · NeurIPS 2019]

Reviewer 1



This work is devoted to repeated (dynamic) contextual auctions with a single strategic buyer. In its setting, knowledge on buyer prior distribution is not available to the seller. The objective of the seller is to find a pricing (policy) that minimize cumulative regret. The key important differences of the considered scenario to previous works are: - The revenue of the seller is compared with respect to a dynamic benchmark; - Specific stochastic dependence of valuations on features of a good. The paper contains a lot of statements, and, thus, seems to be based on a huge work done before. The main issues of the work: 1. Key related work is missed: a. The authors study repeated auction scenario where a buyer is strategic up to a certain accuracy factor. Such buyer behavior has been modeled earlier in [Mohri&Medina, NIPS’2015]. Can the authors position their study with respect to this prior art? b. The authors have references to non-state-of-the-art results. For instance, they discuss the scenario of non-contextual auctions (in Lines 73-77), where optimal solutions with tight regret bound in O(log log T) have been found in [Drutsa, WWW’2017; Drutsa, ICML’2018]. However, the authors provide references to suboptimal algorithms from [Mohri&Medina, NIPS’2014] and [Amin et al, NIPS’2013] only. c. Robust dynamic pricing for repeated contextual auction has been also studied in the work [Cohen,EC’2016]. >Comment after rebuttal: I love the detailed and clear comparison with the missed related work. Please, add this discussion to the next revision of your work. 2. In the introduction, in Lines 38-41: “[Medina & Mohri, 2014] … only provide guarantees against the revenue-optimal static benchmark, which does not take advantage of auction state across time and whose revenue can be arbitrarily smaller than the optimal dynamic benchmark”. It is quite unclear, since in [Medina & Mohri, 2014], the benchmark is the best possible one, because it is equal exactly to the valuation of the buyer and, hence, generate the maximal revenue each round. So, even any dynamic pricing cannot provide higher revenue than this one. The same issue occurs in Lines 81-83. >Comment after rebuttal: I got the answer in general. I hope, the authors will improve clearness in the lines that I have indicated above. 3. The setup in Lines 89-99 is very difficult to follow. For example, reading it three times, I still don’t understand which of the setup is considered: stochastic valuations, or parametric ones, or both simultaneously. >Comment after rebuttal: Thank you for your clarification. I would suggest you to possibly include some scheme of how the workflow of the sampling process looks like and how it relates to the information knowledge both of the buyers and the seller. 4. Most statements in the paper are given without proofs or even their sketches. The proofs are deferred to Appendix. It’d be great to have more insights on the proofs in the main text of the paper. >Comment after rebuttal: Thank you. I hope more pages will allow you to include more insights and intuitions behind their proofs. The key results of the paper rely on the assumption that the seller knows up front that the buyer utilizes linear model to derive his valuation. Does this assumption realistic? Does the results of the study holds for non-linear dependences? >Comment after rebuttal: I got the answer. One suggestion for paper improvement: is it possible to make your algorithms workable in the scenario of kernel models? (see how [Cohen et al.,EC’2016] and [Amin et al, NIPS’2014] easily extended their results from linear models to kernel ones) Also, I noted that there are no conclusions. The paper terminates at Theorem statement. It seems like the paper is not ready for publication in the current version. >Comment after rebuttal: Thank you for your response. I raise the score believing and hoping that you will update the presentation according to your answer. References: Mohri, Mehryar, and Andres Munoz. "Revenue optimization against strategic buyers." Advances in Neural Information Processing Systems. 2015. Cohen, Maxime C., Ilan Lobel, and Renato Paes Leme. "Feature-based Dynamic Pricing." Proceedings of the 2016 ACM Conference on Economics and Computation. ACM, 2016. Drutsa, Alexey. "Horizon-independent optimal pricing in repeated auctions with truthful and strategic buyers." Proceedings of the 26th International Conference on World Wide Web. International World Wide Web Conferences Steering Committee, 2017. Drutsa, Alexey. "Weakly Consistent Optimal Pricing Algorithms in Repeated Posted-Price Auctions with Strategic Buyer." International Conference on Machine Learning. 2018.

Reviewer 2



This paper studies the problem of a revenue maximizing seller setting prices in a repeated dynamic contextual setting for a utility maximizing buyer. The core of the mechanism starts with the NonClairvoyantBalance mechanism in [Mirrokni et al 2018], which is a mixture of 3 auctions: free, Myerson pricing, and posted price w/ fee. In that work, the mechanism is run in each stage based off of the seller's knowledge of buyers distributions, and is parameterized by a "bank balance" that keeps track of the utility accrued over time. One auction is added here - a random posted price, and this auction is used to get the robust non-clairvoyant dynamic mechanism. More discussion would be helpful surrounding the exact benefit from adding the additional auction into the mix. The paper then tackles the contextual dynamic auctions problem in non-clairvoyant environments, combining a variant of the first 4-auction mechanism with a learning policy from Golrezaei et al 2018. In the process, the posted price auction becomes much more intricate (Def D.1, Hybrid Posted Price w/ Extra fee). The exposition covers directly what is happening, but lacks motivation for why things are happening. This combination of approaches builds very materially on prior work. [After response: Thank you, please include in final version]. The paper is generally well written, though more motivation throughout for decisions would be helpful (e.g., the choices of auctions constituting the mechanism - why these four, and why can't we drop any?). [After response - Thank you, please include in final version]

Reviewer 3



The presentation is good overall, but sometimes a bit hard to follow due to the level of technicality and the number of different settings/results. - As a non-expert I had a bit a problem following the presentation, sometimes an intuition could help, e.g. for fee or the balance (it becomes more clear later though). - And sometimes it would be nice to also have a formal proof for prop3.1 (e.g., that (BI) in fact holds). (Even though it may be trivial, it would be nice to see it explicitly.) Similarly, for the two main Theorems of the paper. I was surprised that there are not proofs for them. I guess it trivially follows form the lemmas but please (easily) save the reader some time here. - This being said, the paper overall, incl. supplement, looks rather good and polished. In terms of originality/significance, they do seem to strongly build on previous work in this direction, but seem to have enough interesting own contributions. - In particular, from how it looks, they build on Mirrokni [2019]. Mirrokni also treats non-clairvoyant mechanisms (i.e., future type (=valuation) distributions are unknown), but Mirrokni does not *learn* the (current) type distribution. In terms of quality, in the proofs they seem to be knowing what they are doing, but I didn't check the details. Looks interesting though. This is a purely theoretical paper without experiments but this is fine with me. After author response: Thanks for the response!

[Author Response · NeurIPS 2019]

We thank all the reviewers for their detailed and insightful feedback.

**Reviewer 1:** We thank the reviewer for the detailed feedback and pointers to missing literature.

1. We would like to clarify that the buyer in our scenario is not just "strategic up to a certain accuracy factor". Our buyer
is an exact utility maximizer whose objective is to maximize her long-term cumulative utility. This is different from the
$\epsilon$-strategic buyer proposed in [Mohri and Medina, 2015], which allows for $\epsilon$-suboptimal responses. In our work, the
notion of $\eta$-DIC is a property of our mechanism, not an assumption on the buyer's behavior. The property ensures that a
utility maximizer will always report bids close to her true valuations (i.e., within a bounded additive error).

2. An important distinction with [Drutsa, 2017, 2018] is that our work focuses on stochastic settings in which the
buyers' valuations are redrawn independently at each round, whereas their work considers a fixed valuation model. In
the stochastic setting, the revenue gap between the optimal dynamic mechanism and the optimal static mechanism
could be arbitrarily large [Papadimitriou et al., 2016], while there is no difference for the fixed valuation model. As for
[Cohen et al., 2016], they consider robust pricing in repeated contextual auctions for myopic buyers, while we consider
a buyer who aims to maximize her long-term utility. We agree, however, that we should improve the coverage of the
fixed-valuation model and the existing works on robust pricing in the related work. We will include the citations you
brought up and add a discussion of optimal regret bounds under fixed valuations in the revision.

3. Our model of contextual auctions is both stochastic and parametric: The buyer's valuation distribution is characterized
by her fixed but private preference vector, while the features of the items and the distributions of market noise at each
stage are stochastic. Therefore the buyer's realized value at each stage is stochastic. This directly follows and extends
the contextual auction model proposed in the literature [Amin et al., 2014, Golrezaei et al., 2018].

4. We would have loved to discuss more in the main body. But given the technical nature of our proofs and the 8-page
limit, we had to move most proofs to Appendix. We will make sure to improve our presentation by providing more
high-level ideas and proof sketches, and make space for conclusions in the revision.

5. Our robust non-clairvoyant mechanism only relies on Assumption 2 and does not depend on the parametric valuation
model. Our no-regret policy for contextual auctions does depend on the valuation model. It would be an interesting
avenue for future work to consider nonlinear valuation models or even non-parametric models. We would like to
emphasize though that we view the robust non-clairvoyant mechanism as the most important contribution, as it opens
the possibility to take an existing robust pricing mechanism for contextual auctions from [Golrezaei et al., 2018] (which
is no-regret against the optimal static benchmark) to obtain a mechanism with no-regret against the optimal dynamic
benchmark (albeit with several technical updates), rather than just the optimal static benchmark.

6. We do not have a matching lower bound, and we think it is an important open question to improve the regret guarantee
or to provide a matching lower bound. We consider the upper bound a good advancement since there was no previous
no-regret bound against the optimal dynamic benchmark in our setting.

Given that these were Reviewer 1's main concerns, we sincerely hope Reviewer 1 would consider revising their score.

**Reviewer 2:** We thank the reviewer for the positive and encouraging review.

1. Why these four mechanisms: The give-for-free mechanism, the posted-price auction with fee, and the Myerson's
auction are used to guarantee a $\frac{1}{3}$ approximation against the dynamic benchmark (when the distributional information is
perfect), while the random posted price auction is added to obtain robustness.

2. Intuition behind the hybrid mechanism: The hybrid mechanism uses static stage mechanisms for stages with small $a_t$
and dynamic stage mechanisms for stages with large $a_t$. Intuitively, for the static mechanisms, we first use give-for-free
mechanisms for the first few items with small $a_t$ in each phase to accrue a large enough bank account balance for
the buyer. Later, this allows us to almost always implement a give-for-free mechanism with extra fee $\mathbb{E}[v_t]$ for later
stages with small $a_t$ to extract full welfare. The key observation is that the dynamics of the bank account balance
in give-for-free mechanisms with extra fee is a martingale and therefore, starting with a large enough balance, the
probability that the balance becomes close to 0 is small. For stages with large $a_t$, we apply the same dynamic stage
mechanisms as the non-clairvoyant mechanism proposed in Mirrokni et al. [2018]. Finally, both the static and dynamic
stage mechanisms are made robust using our framework, by mixing in the random posted-price auction.

3. Assumption 1 is only used in the analysis of our no-regret policy and our robust non-clairvoyant mechanism only
depends on Assumption 2. Without Assumption 1, simply consider a worst case scenario in which $a_1 = T$ while $a_t = 1$
for all $t > 1$. In this case, the revenue loss could be $\Omega(T)$ from the first stage since the seller has no information to
estimate the buyer's private preference at the first stage.

**Reviewer 3:** We thank the reviewer for the positive and encouraging review. We will make sure to improve our
presentation by providing more high-level ideas and proof sketches, and make space for conclusions in the revision.

[Meta-Review · NeurIPS 2019]

Repeated auctions with strategic agents is an interesting and hot topic. This paper generalizes a previous one where some value distribution is now learned on the fly - instead of being known beforehand (as in online learning). The reviewers pointed out the merits of that paper (the aforementioned contribution) which is worth acceptance at NeurIPS this year. One reviewer had some concerns about the contents of the paper, but they disappeared after the rebuttal and the discussion.